# High-throughput characterization of bacterial responses to complex mixtures of chemical pollutants

**Thomas P. Smith** ● ✉**, Tom Clegg** ● **, Emma Ransome, Thomas Martin-Lilley, James Rosindell, Guy Woodward, Samraat Pawar** ● **& Thomas Bell** ●

Our understanding of how microbes respond to micropollutants, such as pesticides, is almost wholly based on single-species responses to individual chemicals. However, in natural environments, microbes experience multiple pollutants simultaneously. Here we perform a matrix of multi-stressor experiments by assaying the growth of model and non-model strains of bacteria in all 255 combinations of 8 chemical stressors (antibiotics, herbicides, fungicides and pesticides). We found that bacterial strains responded in different ways to stressor mixtures, which could not be predicted simply from their phylogenetic relatedness. Increasingly complex chemical mixtures were both more likely to negatively impact bacterial growth in monoculture and more likely to reveal net interactive effects. A mixed co-culture of strains proved more resilient to increasingly complex mixtures and revealed fewer interactions in the growth response. These results show predictability in microbial population responses to chemical stressors and could increase the utility of next-generation eco-toxicological assays.

Natural environments are under increasing pressure from multiple anthropogenic stressors[1,2], and freshwater systems are no exception. Freshwater environments are increasingly exposed to toxic chemical pollutants at local to global scales[3,4], raising substantial concerns for ecosystem health[5]. Understanding the effects of chemical pollutants on natural systems is therefore key to understanding ecosystem health. A particularly important aspect is to understand how chemical pollutants affect the microbes embedded within ecosystems. Microbes are globally ubiquitous drivers of key ecosystem processes and services in their roles as decomposers, mutualists, food sources, chemical engineers and pathogens[6,7]. As such, stressor impacts on microbes can ripple through the wider ecosystem, with changes in these largely overlooked taxa potentially altering the functioning of entire food webs[8,9].

The huge scope for interactions among complex chemical mixtures generates a large amount of uncertainty when extrapolating classical ecotoxicological studies to ecosystem processes[10–12].

In combination, stressors may produce effects on biological systems that are equal to, stronger than (synergistic) or weaker than (antagonistic) the sum of their parts[10]. Pollution mitigation strategies may therefore produce unexpected or even detrimental effects on ecosystem processes when synergies or antagonisms among chemical stressors are not well understood, so a more complete understanding of stressor interactions is required[12]. Natural systems are typically subjected to complex cocktails of chemical stressors, but the importance of multi-way interactions among three or more stressors ('higher-order interactions') remains largely unknown[13]. Data are a key limitation because testing for higher-order interactions requires information on not only the effect of multiple stressors acting together but also the effects of every subset of those stressors[13,14]. If higher-order interactions are common, it would present a substantial challenge for understanding and predicting the effects of chemical stressors in the real world.

The Georgina Mace Centre for the Living Planet, Department of Life Sciences, Silwood Park Campus, Imperial College London, Ascot, UK.
✉e-mail: thomas.smith1@imperial.ac.uk

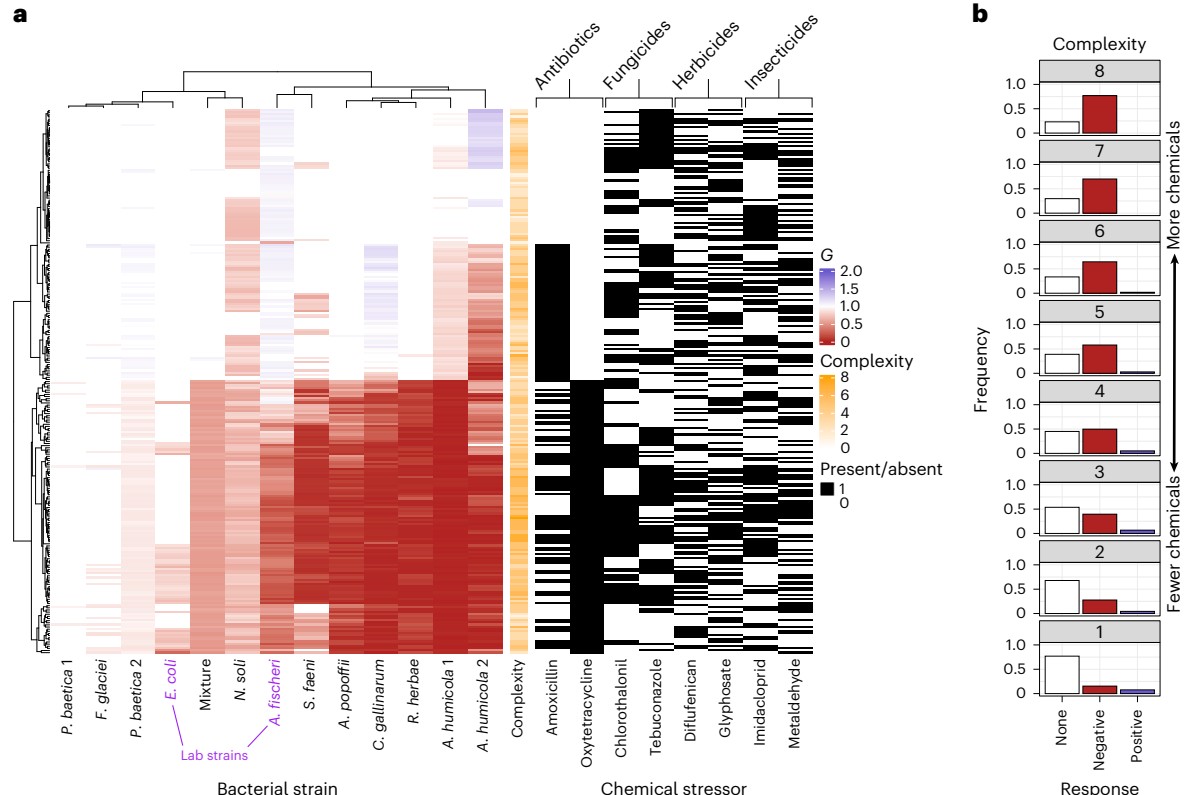

**Fig. 1 | Chemical stressor responses vary between strains. a**, Heat map of all chemical stressor responses. Left: the mean ($n$ = 4 biologically independent replicates) relative growth ($G$) in the presence of chemical stressor(s) on a scale from positive responses (blue) to to negative responses (red). Colours are only shown where $G$ is significantly different from 1 (that is, the chemical mixture has a significant impact on growth). Each column is the fingerprint of responses for a given strain; each row is a particular mixture of chemicals. Columns are clustered by similarity between strains; rows are clustered by similarity between responses.

Each chemical stressor present in a given mixture is indicated by black lines in the right panel. The number of chemicals in a mixture ('complexity') is shown in orange. Model strains are identified in purple; the mixture of strains is highlighted in bold. Chemicals are grouped by their target organisms. **b**, Proportion of strain and chemical mixtures showing negative, positive or no growth response, grouped by the number of chemicals in the mixture. Most chemicals alone have no impact on most bacteria; however, increasingly complex mixtures of chemicals have increasingly negative effects on growth.

Another limitation of current ecotoxicology assays is the lack of evidence for their applicability beyond a narrow focus on a tiny number of 'model' organisms or lab strains (for example, *Aliivibrio fischeri*[15] and *Escherichia coli*[16]), which may bear little resemblance to naturally occurring biota. To gain a more general understanding of the impacts of chemical stressors on bacteria, we must therefore expand ecotoxicological testing to include non-model species that are more representative of natural microbial communities.

In this article, we assayed population growth across a diverse set of bacterial taxa to quantify their responses to mixtures of chemical stressors known to pollute natural environments. By testing the responses of both model and non-model strains of bacteria, we asked whether these responses were phylogenetically conserved and thus generalizable through evolutionary relatedness. Our high-throughput assays allowed us to thoroughly investigate the importance of high-order interactions in complex chemical mixtures.

## Results

### Chemical stressor responses vary among bacterial strains

We assayed the growth of two model strains of bacteria (*A. fischeri* and *E. coli*), ten environmental strains (*Aeromonas popoffii*, *Carnobacterium gallinarum*, *Flavobacterium glaciei*, *Neobacillus soli*, *Rhizobium herbae*, *Sphingomonas faeni*, two strains of *Arthrobacter humicola* and two strains of *Pseudomonas baetica*; see Supplementary Table 2 for details) and a mixed culture of all ten environmental strains (Methods). The environmental strains were isolated from pristine freshwater systems in Iceland with no history of chemical exposure and therefore considered naive to chemical stress. We quantified their growth in 255 chemical stressor mixtures (every possible combination of 8 stressors), as well as under control conditions (no stressors). These stressors are eight different chemical pollutants known to be prevalent in freshwater environments, targeting four components of freshwater ecosystems (bacteria, fungi, plants and invertebrates). We quantified growth as the area under the bacterial growth curve (AUC) and calculated the effect of a chemical mixture as the growth relative to growth under control conditions ($G$); see Supplementary Fig. 1 for the responses of bacteria to each chemical. The growth responses of all bacteria–chemical combinations are visualized in Fig. 1. Across all bacteria tested, the chemical mixtures produced increasingly negative effects on growth as the number of chemicals in the mixture increased (Fig. 1b; linear regression of frequency of negative responses against number of chemicals in mixture: intercept = 0.11, slope = 0.09, $P < 0.001$, $r^2 = 0.98$). We used hierarchical clustering to group these responses of bacteria to each specific chemical mixture both by similarity between bacterial strains and by similarity between chemical mixtures (Fig. 1a). The chemical mixture responses clustered into two groups—those mixtures with and those without oxytetracycline (Fig. 1a). We also observed strong clustering for amoxicillin in the absence of oxytetracycline and some clustering of responses to both fungicides (chlorothalonil and tebuconazole) in both the presence and absence of oxytetracycline. Responses to mixtures containing the same numbers of chemicals did not generally cluster together; that is, the similarity of responses across strains was driven by the presence of specific chemicals, rather than by the number of chemicals.

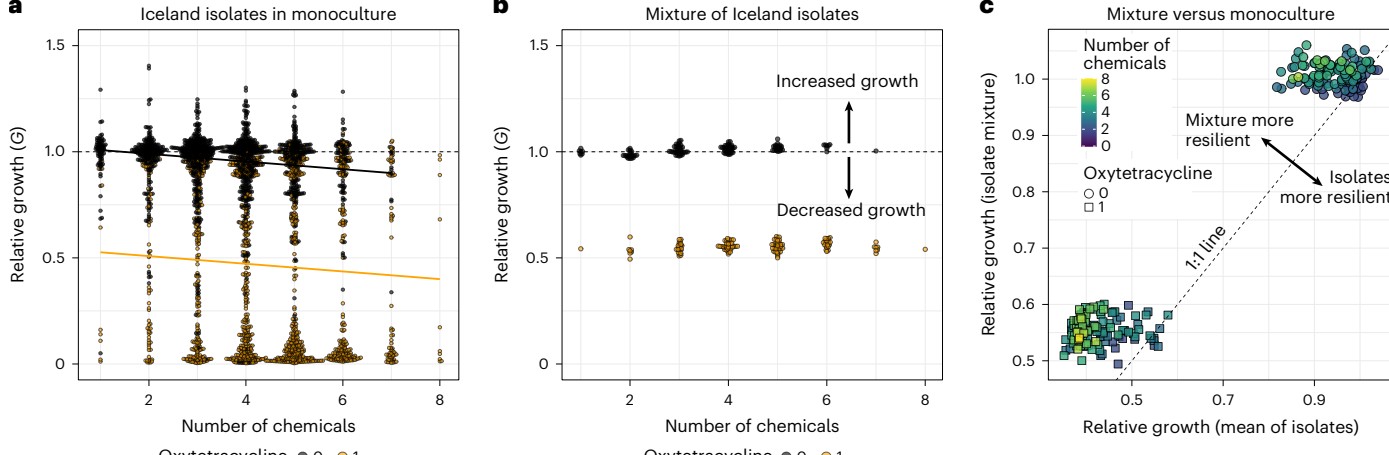

**Fig. 2 | Mixtures of increasing numbers of chemical stressors reduce bacterial growth in monoculture but not in community. a**, Combined responses of all environmental bacterial strains to all chemical mixtures, given as growth (area under the growth curve) in chemical mixture relative to control growth for a given strain ($G$). Dashed line marks 1, below which there is reduced growth and above which there is increased growth. There is bimodality in the responses, with mixtures containing oxytetracycline (orange) showing lower growth on average than those without (black). As the number of chemicals in the mixture increases, growth is on average reduced across strains, both in the presence and absence of oxytetracycline (linear regression, intercept = 1.02, slope = −0.03, oxytetracycline presence = −0.42, $P < 2.2 \times 10^{-16}$, $r^2 = 0.33$). **b**, Responses of the mixture of environmental strains to all chemical mixtures. There is similar bimodality to the monocultures, due to the presence of oxytetracycline; however, the addition of further chemicals has an almost negligibly small impact on growth (linear regression, intercept = 0.99, slope = 0.005, oxytetracycline presence = −0.46, $P < 2.2 \times 10^{-16}$, $r^2 = 0.99$). **c**, The mean growth response for the Iceland isolates in monoculture ($x$ axis) against the growth response of the mixed culture ($y$ axis). Points are coloured by the number of chemicals present. Squares are mixtures containing oxytetracycline. Below or above the 1:1 line, the mixed culture shows lower or higher relative growth than the mean of isolates in monoculture, respectively. The mixed culture is more resilient to the negative growth effects of increasingly complex chemical mixtures than the mean of the isolates in monoculture. In all plots, points are a mean of four replicates.

To understand the biotic context of chemical impacts, we compared the responses of the environmental strains in monoculture to their responses in a mixed co-culture (Fig. 2). In monoculture we observed a decrease in growth on average as the strains were exposed to mixtures of sequentially more chemicals (Fig. 2a). However, the distribution of responses was bimodal, with many chemical combinations producing either a very weak effect or a very strong effect, depending on the presence or absence of the antibiotic oxytetracycline (Fig. 2a; linear regression, intercept = 1.02, slope = −0.03, oxytetracycline presence = −0.42, $P < 0.001$, $r^2 = 0.33$). These responses were also highly variable among strains, with some bacteria strongly impacted but others resilient to the chemicals used here (Fig. 1a and Supplementary Fig. 2). In the mixed culture of strains, we observed the same bimodality of responses in the presence and absence of oxytetracycline, but by contrast, the addition of more chemicals had little impact on growth (Fig. 2b; linear regression, intercept = 0.99, slope = 0.005, oxytetracycline presence = −0.46, $P < 0.001$, $r^2 = 0.99$). This bimodality in responses echoes the strong clustering of responses to oxytetracycline seen in Fig. 1. Comparing the mean response of strains in monoculture to the realized response of strains in mixed culture, we show that the mixed culture of bacteria is more resilient to the negative impacts of the chemical treatments than predicted by the monoculture responses (Fig. 2c).

Replicated species from different locations, *Pseudomonas baetica* 1 and 2 and *Arthrobacter humicola* 1 and 2, showed similar responses to the chemical mixtures (Fig. 1a). We tested for phylogenetic signal in these chemical responses by computing the pair-wise distance matrix from the stressor responses and comparing this to the distance matrix from the phylogeny of the strains, constructed from their 16S sequences (Fig. 3a and Supplementary Fig. 3). Using a Mantel test based on Kendall's rank correlation $\tau$, we found no significant correlation between the chemical responses and phylogeny; that is, the chemical responses are not generalizable by evolutionary relatedness ($\tau = 0.076$, significance = 0.154). To ensure that these findings were not being driven by the strong impact of oxytetracycline, we repeated the analysis without oxytetracycline and obtained qualitatively the same result ($\tau = 0.049$, significance = 0.322; Supplementary Fig. 4). To further test for evolutionary signal in the growth responses of bacteria to these chemicals, we calculated Pagel's $\lambda$ and Blomberg's $K$, two metrics of phylogenetic signal. The responses to amoxicillin show a strong signal of phylogenetic heritability ($\lambda = 0.82$, $P = 0.04$; $K = 0.89$, $P = 0.03$); however, the responses to all other chemicals show much weaker, non-significant signal for both metrics (Fig. 3b); that is, for the majority of these chemicals, shared evolutionary history does not drive the distribution of growth responses at the tips of the phylogeny. See Supplementary Table 1 for full results of these tests.

**Testing interactions within chemical mixtures**

We quantified two measures of the structure of interactive effects of multiple chemicals on both the single-population and mixed culture growth trajectories. 'Net' interactions quantified the overall interaction among the stressors in a mixture, without disentangling specific interactions between stressors. 'Emergent' interactions quantified specific higher-order interactions among multiple stressors. We developed a methodology for quantifying these two measures of interactions by comparing the growth in mixtures with the expectations under a multiplicative null model (Fig. 4a,d) and then tested for significance via bootstrapping (see Methods for full details). We categorized significant interactions as antagonistic (chemicals dampen the effects of each other) or synergistic (chemicals amplify the effects of each other) if the response was weaker or stronger than predicted by the null model, respectively (Methods). If no significant interaction was found, the response is categorized here as multiplicative; that is, the chemical mixture produced a growth response significantly different from the control growth, but this response was equivalent to the multiplicative combination of growth responses to each individual chemical present in the mixture. We found that across all strains in monoculture, 16% of two-way chemical mixtures produced a significant interaction (there

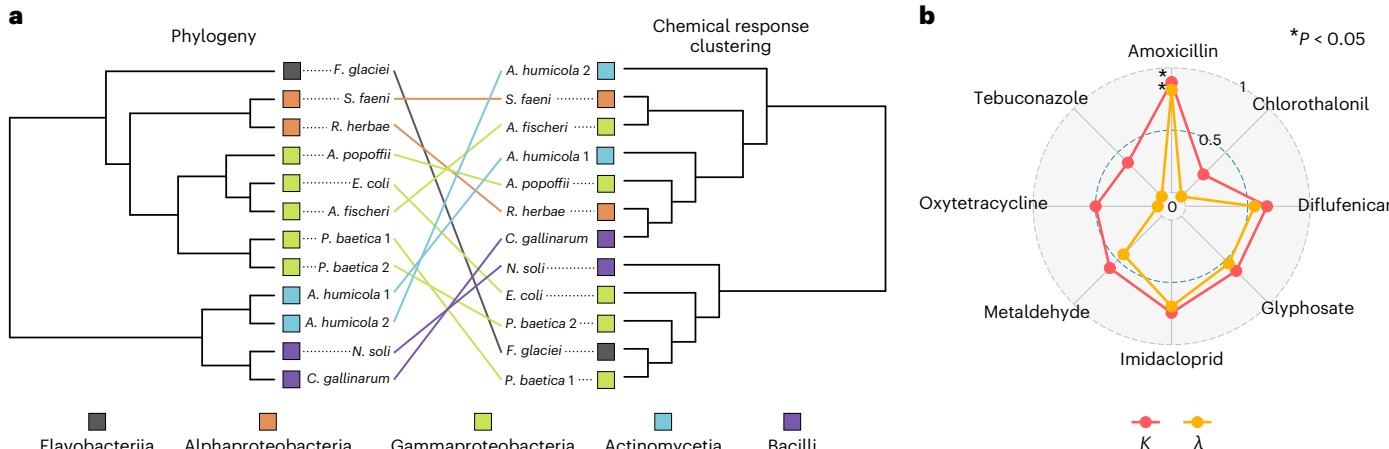

**Fig. 3 | Chemical responses are not strongly driven by phylogenetic relatedness. a**, Phylogeny of the bacterial isolates from their 16S sequences juxtaposed against their clustered phenotypic distances based on growth responses to the chemical stressors. These are presented as cladograms for visualization purposes, and thus branch lengths are not indicative of phylogenetic or phenotypic similarity. Strains are coloured by the their taxonomic class, and lines show placement of the same species in the phylogeny and phenotypic clustering. We find no significant correlation between the phylogenetic and phenotypic distance matrices (one-sided Mantel test,

$P = 0.154$). **b**, Testing for phylogenetic signal in the responses of all strains to each single-chemical treatment, using Blomberg's $K$ and Pagel's $\lambda$ metrics. Briefly, values close to 0 indicate no phylogenetic signal, values of 1 approximate a Brownian motion model of trait evolution; see Methods for full details. $P$ values for $\lambda$ are derived from a log-likelihood ratio test; $P$ values for $K$ are derived from a randomization test ($n = 1,000$)[53]. Significant phylogenetic signal ($P < 0.05$) indicated by an asterisk. $P$ values are not adjusted for multiple comparisons. Only the responses to amoxicillin show strong phylogenetic signal ($K$, $P = 0.029$; $\lambda$, $P = 0.037$). See Supplementary Table 1 for all $P$ values and test statistics.

is no distinction between net and emergent interactions in mixtures of two chemicals). As the number of stressors increased, proportionally more treatments showed a net interaction (Fig. 4b). However, significant emergent interactions were proportionally less common for the monocultures in more complex chemical mixtures (Fig. 4e). For half of the 12 strains, the eight-way mixture showed an overall net interaction, but in no cases did we detect any significant seven- or eight-way emergent interactions. By comparison, we found very few interactions in the mixed culture of isolates (Fig. 4c,f), and while two net interactions did persist into six-chemical mixtures (Fig. 4c), no new emergent interactions arose in mixtures of more than three chemicals (Fig. 4f). Essentially, we cannot rely on the responses of single species of bacteria to represent the responses of whole microbial communities to chemical mixtures.

Across all bacterial strains tested in monoculture, in most chemical mixtures we found no interactions: when testing for net interactions, either there was no response or the multiplicative null model was favoured in 68% of all mixtures, whereas when testing for emergent interactions either there was no response or the multiplicative null model was favoured in 91% of all mixtures. Where significant interactions were found in monoculture, antagonisms were more common than synergisms (Fig. 4b,e). In the two-chemical mixtures, we found 46 antagonisms and 8 synergies (85% antagonistic). In more complex mixtures (3 or more chemicals), we found 601 net antagonisms and 284 net synergies overall (68% antagonistic). By comparison, in these more complex mixtures, we found 170 emergent antagonisms and 31 emergent synergies (85% antagonistic). Furthermore, these higher-order emergent interactions were increasingly likely to be antagonistic in more complex mixtures (significant 3-chemical emergent interactions = 64% antagonistic; 4 = 94%; 5 = 98%; 6 = 100%). In the mixed culture of pristine strains, there were an equal number of antagonisms and synergies in the two-chemical mixtures. However, all interactions in higher-complexity mixtures (3 or more chemicals) were antagonistic (Fig. 4c,f). The mixed culture was therefore similar to the monocultures in that antagonisms were generally the more prevalent interaction observed.

The significant net interactions that we detected were not consistent in their qualitative effect across bacterial strains, with certain

strains showing synergistic and others experiencing antagonistic interaction effects for the same mixture (Fig. 5). Although fewer in number, emergent interactions were more consistent among strains than net interactions—where multiple strains showed an emergent interaction for a particular mixture, in almost all cases this interaction was of the same type (that is, synergistic or antagonistic). Specifically, in 86% of mixtures of three or more chemicals where multiple strains had an emergent interaction, the same interaction type manifested across all strains (Supplementary Fig. 5). Where lower-level interactions were evident, these often persisted in more complex mixtures containing the interaction-producing subset, leading to the same overall net interaction (Fig. 5). Interactions in mixtures containing oxytetracycline (antibiotic) and tebuconazole (fungicide) were particularly common (Fig. 5 and Supplementary Fig. 6). In higher-complexity mixtures containing both oxytetracycline and tebuconazole, net interactions of the same type as the two-way interaction (antagonistic or synergistic) were also likely to be observed (Supplementary Fig. 6). Across all strains, the most prevalent higher-order emergent interactions occurred in mixtures containing both oxytetracycline and metaldehyde (Supplementary Notes and Supplementary Fig. 7).

To test how many interaction terms were required to explain the net responses at different levels of chemical mixture complexities, we quantified interactions using null models incorporating sequentially higher levels of interaction terms. We found that the more complex mixtures required more interaction terms to explain their effects. However, incorporating only two- and three-way interactions into the models was sufficient to explain 50% of the net interactions at all levels of mixture complexity (Table 1). We can therefore conclude that the majority of interactive effects are captured at relatively low chemical diversity.

## Discussion

By systematically assaying bacterial growth in every possible combination of eight chemical stressors, we uncovered interactions in the responses of bacteria to these chemicals. Across all strains and stressor mixtures, we found an increased prevalence of net interactions in more complex chemical mixtures. Most of those net interactions were antagonistic (the responses of bacteria to combinations stressors were dampened compared to their responses to chemicals

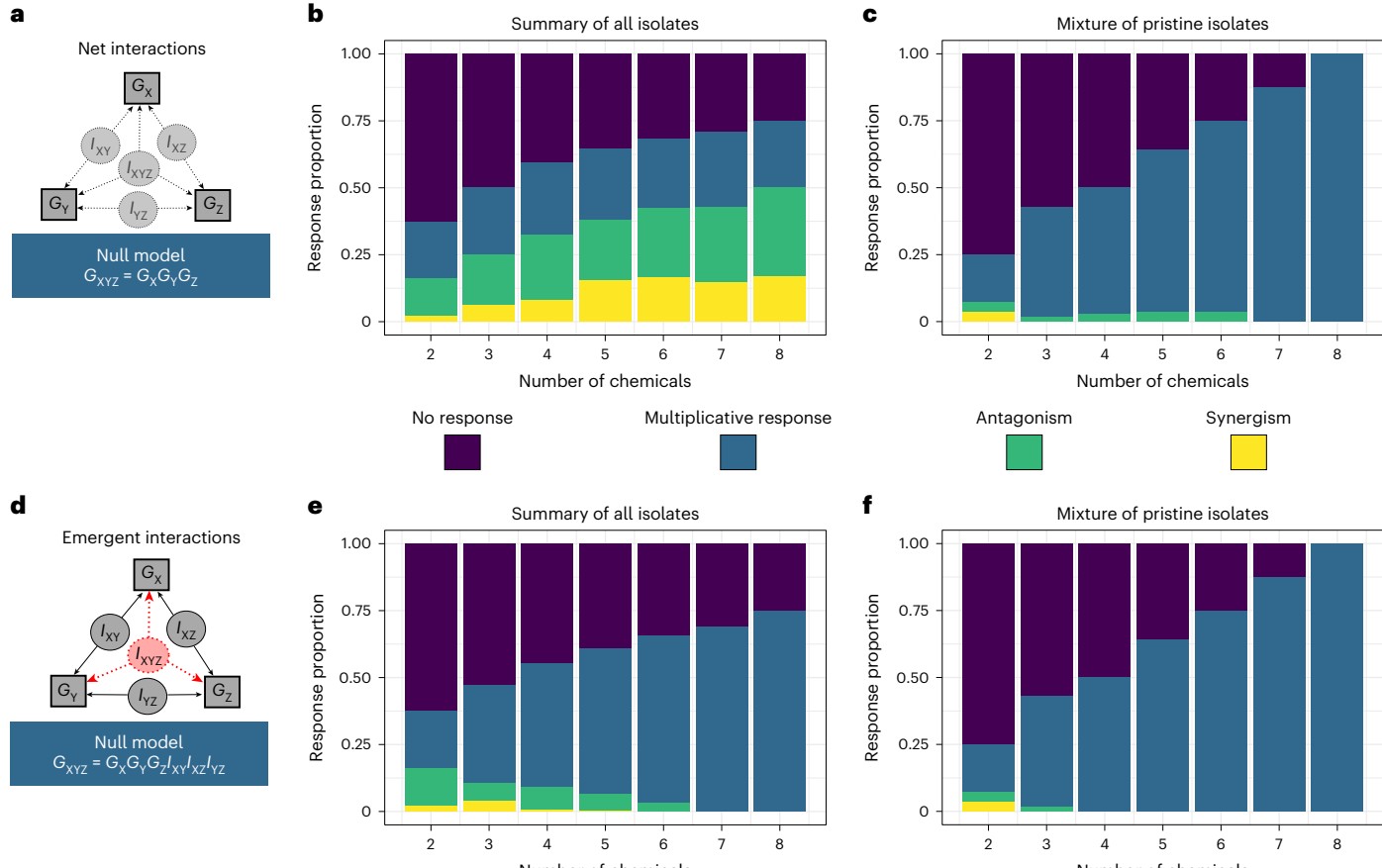

**Fig. 4 | Quantifying multi-stressor interactions. a**, We tested for net interactions in chemical stressor mixtures by quantifying bacterial growth ($G$) in mixture and comparing this to growth in the presence of each stressor (here X, Y and Z) individually. A net interaction can be caused by any of the specific interactions ($I$) that occur within a mixture. For the three-chemical example, this includes all possible two-way interactions as well as the three-way emergent interaction, and thus the null model contains terms for the individual stressor responses only. **b,c**, Bars summarize the proportion of occurrences of each net interaction type (multiplicative response, blue; antagonistic, green; synergistic, yellow; no response, purple) for every mixture of a given number of chemicals, for every strain of bacteria in monoculture (**b**) and the mixed culture (**c**). **d**, We test for emergent interactions (here the three-way interaction, red) using a null model which accounts for all lower-order terms. **e,f**, The proportion of occurrences of emergent interactions for all monoculture isolates (**e**) and the mixed culture (**f**).

individually); however, the relative prevalence of synergistic net interactions increased in more complex chemical mixtures. Conversely, we found fewer emergent (higher-order) interactions in more complex mixtures, but these were increasingly more likely to be antagonistic in higher-chemical-complexity mixtures. These results are consistent with recent work that show emergent antagonisms in mixtures with higher numbers of stressors but also increased frequencies of net synergies[13,17–19]. This work extends our understanding of stressor responses in both the biological context (non-model bacteria and a mixed culture) and abiotic context (complex mixtures of environmental pollutants), paving the way for future work to link the effects of microbial stressor responses with ecosystem processes.

Understanding the importance of interactions at different levels of mixture complexity is a key limitation for predicting the impacts of multiple stressors[10–12]. We found that most net interactions could be explained by two- and three-way interactions, rather than by higher-complexity interactions among stressors. That is, lower-order effects persist in more complex mixtures of chemicals, resulting in these net interactions. This is in general agreement with work on the responses of bacteria to multi-drug combinations which has established that the effects of drug pairs are often sufficient to infer the effects of larger combinations[20]. That the net effects of combinations of many stressors can be predicted by incorporating relatively few of all the potential underlying interactions simplifies the challenge of

predicting the responses of microbial populations and communities to multiple chemical stressors. Recent meta-analyses are consistent with our experimental results in showing that antagonisms are the prevalent stressor interaction type at organism, population and community levels in freshwater ecosystems[21–23]; however, these meta-analyses have generally ignored the microbiota and do not show causality. Our high-resolution microbial experiments overcome these key limitations.

The results have implications for the responses of natural bacterial communities, and their associated ecosystem processes, to chemical mixtures. We found that the growth of a mixed culture of strains both was less impacted by the addition of multiple chemicals and showed many fewer interactions than expected from the growth of the strains in monoculture. When chemicals inhibit part of the community, compensation by resistant taxa may rescue important functions, such that growth is maintained. However, while the growth of microbial communities may be resilient to multiple stressors in a broad sense, interactions may still be impactful in natural communities. In particular, keystone taxa can have a disproportionate impact on community structure and function[24,25]. At a single strain level, we found highly variable responses to the chemical mixtures. If unique interactions lead to the loss of keystone species, this could have comparatively large effects on community functioning and stability. In communities, biotic interactions among species may also affect stressor responses[26–28]. In particular, differing levels of competition and facilitation can drive

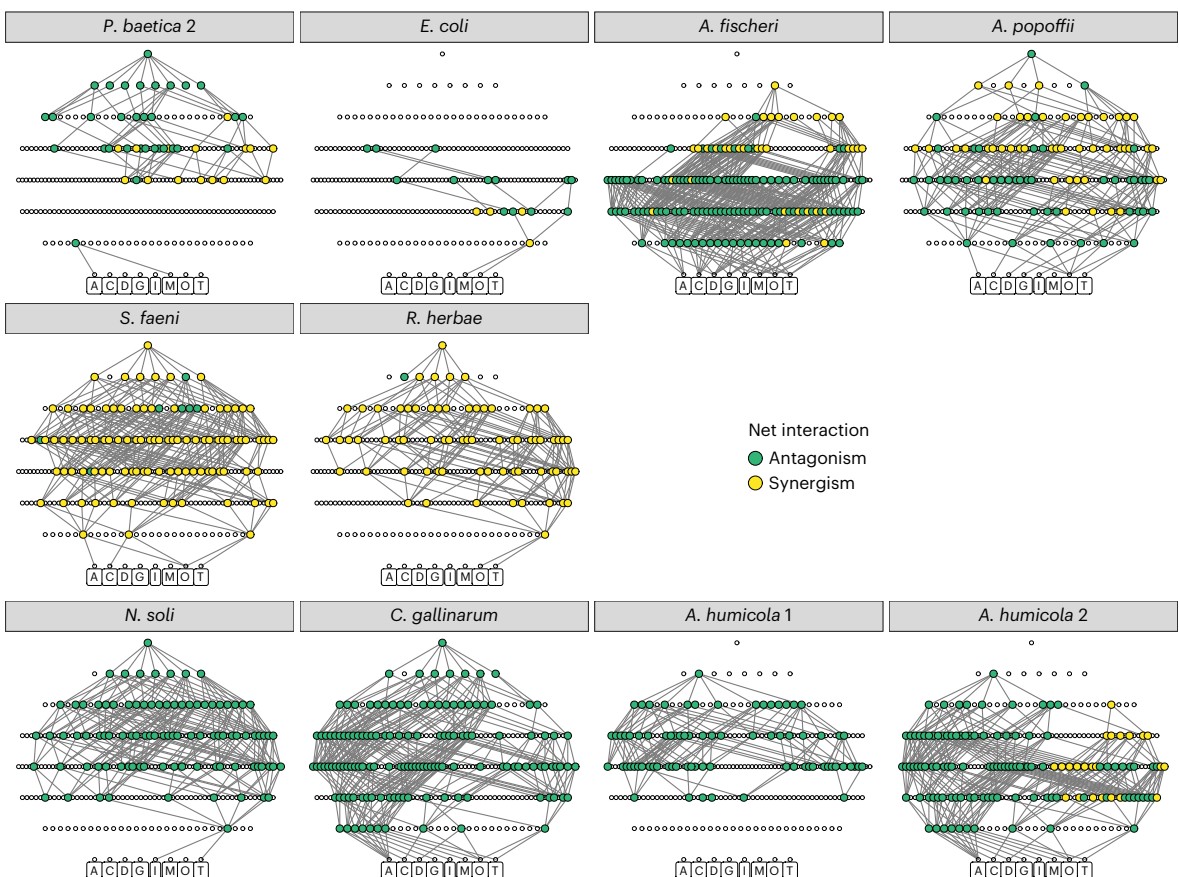

**Fig. 5 | Lower-level interactions persist in higher-complexity chemical mixtures.** Net interactions visualized as networks for each strain. Each point represents a different chemical mixture with the bottom row representing each individual chemical (designated by the first character of their name below the point) and every subsequent row above being a more complex mixture of these chemicals, finishing with a single point for the eight-chemical mixture. Nodes without a significant interaction are left as unfilled circles; nodes with interactions are larger and coloured by antagonism (teal) or synergism (yellow). Edges are drawn between nodes with significant interactions one row apart where the mixture below is a subset of the mixture above. Strains *P. baetica* 1 and *F. glaciei* are omitted from this figure due to a lack of interactions to visualize (*P. baetica* no interactions, *F. glaciei* a single net interaction). Networks are ordered here based on the phylogeny (Fig. 3a).

**Table 1 | The number of interaction terms required to explain net interactions in mixtures with different numbers of chemicals (complexity)**

| Mixture complexity | Interactions included in model | | | | | |
|---|---|---|---|---|---|---|
| | One-way (net effects) | Two-way | Three-way | Four-way | Five-way | Six-way |
| 3 | 169 | 49 (29%) | | | | |
| 4 | 272 | 112 (41%) | 58 (21%) | | | |
| 5 | 254 | 131 (52%) | 79 (31%) | 32 (13%) | | |
| 6 | 143 | 77 (54%) | 57 (40%) | 26 (18%) | 6 (4%) | |
| 7 | 41 | 20 (49%) | 18 (44%) | 11 (27%) | 6 (15%) | 0 (0%) |
| 8 | 6 | 3 (50%) | 3 (50%) | 3 (50%) | 2 (33%) | 0 (0%) |

Here we show the number of mixtures across all strains showing a net interaction effect, based on how many levels of interaction terms are incorporated into the null model. The percentage is the number of interactions remaining unexplained by the null model. Across all strains, there are 169 three-chemical mixtures showing a net interaction effect. When including two-way interactions, this is reduced to only 49 mixtures with significant effects (29%); that is, the remaining 49 mixtures require a further interaction term to explain their effect (three-way emergent interaction).

variation in the responses of communities to stress[27,28]. Hence, there is a need for future work to build upon these results by focusing on whole microbial communities, as these are more representative of how chemical stress is encountered in the environment. Community-level data may require different frameworks and definitions of stressor interactions than those used in population-level studies such as ours. For example, an interaction type not explicitly tested here is dominance, where the combined response of stressors may be explained by the effect of one stressor alone[23,29,30]. At a population level, dominance may be considered a special case of antagonism (that is, if one stressor blocks the action of another); however, at a community level dominance may occur due to species interactions or compensation by tolerant species[23].

We uncovered general patterns but found that neither the prevalence and type of interactions nor the overall responses of growth in chemical mixtures were consistent between different strains of bacteria. In part, this may be due to the dose-dependent nature of chemical responses—if a strain is resistant to a chemical mixture at a

given dosage, then there will be no physiological response from which to infer interactions. Thus, increased concentrations of chemical stressors may reveal more interactions in strains that appeared to be insensitive using assays conducted at a single dosage. Bioavailability also plays a key role in this context, because measured concentrations may not reflect the capacity for entry into target organisms[31]. The varied bacterial responses to these mixtures could not generally be predicted by evolutionary (phylogenetic) relatedness. There was also a lack of strong phylogenetic signal in the responses to most chemicals in isolation, with the exception of amoxicillin. This has important implications for ecotoxicology studies that focus on testing single strains of bacteria; it means that most toxicity tests cannot be generalized even to groups of bacteria phylogenetically similar to the strain tested.

Interactions between antimicrobial compounds are often determined by the cellular targets of those compounds (for example, cell wall, DNA replication machinery, ribosome synthesis and so on), with certain target classes leading to specific interaction types with others[32]. A relevant example is bactericidal antibiotics, which require cell growth, and so compounds that inhibit growth can interact antagonistically, supressing the effect of the antibiotic[33]. Our study includes the bactericidal amoxicillin (inhibits cell wall synthesis) and bacteriostatic oxytetracycline (inhibits protein synthesis), and four strains of bacteria in our study do show an antagonism for this pairing. Moreover, the prevalence of other antagonisms observed here in mixtures with oxytetracycline may be due to bactericidal activity of pesticides, inhibiting growth and therefore reducing the efficacy of oxytetracycline. However, there can be a multitude of additional reasons for interactions to arise, including uptake effects, for example, if the cell permeability of one compound is affected by another[32]; if the solubility of one compound is affected by another (potentially relevant to our liquid-culture experiments); direct physical interactions, for example, modes of action sharing the same binding site[32]; and other, more biochemically complex mechanisms. Further work to characterize the mode of action of pesticides on non-target organisms is necessary. This may include investigating the expression of genes related to biochemical responses and in particular should focus on whether specific classes of pesticide compounds affect microbes in similar ways and are thus generalizable. Further work on the mechanistic basis for the interactions we observed would lay the foundation for future ecotoxicology frameworks that predict the impacts of multiple chemical compounds on the non-target microbial components of ecosystems more effectively.

In this study, we moved away from the traditional reliance on a few model bacteria to consider environmental bacteria naive to chemical stress. We tested the effects of an array of pollutants that enter freshwater environments and have been identified as key targets for microbial studies[34]. Our results are therefore relevant to natural systems beyond the laboratory and show that microbes exhibit a similar prevalence of antagonistic and synergistic responses compared to studies of macro-organisms. Understanding the pervasiveness and importance of such interactions is a key challenge faced in predicting multiple stressor impacts for ecosystem management[10]. It is therefore encouraging that patterns in the data are not being driven by highly complex interactions, which would make prediction challenging. It is also encouraging that the growth of a mixed culture of strains was both resilient to multiple chemical stressors and exhibited few complex interactions, indicating that natural microbial communities may be more resilient to multiple stressors than predicted by single-strain experiments. However, the broader impacts on ecosystem functioning remain untested, and we cannot rule out important impacts on keystone species. The high level of variability in the responses of different bacterial strains shows the potential for harnessing bacteria as high-resolution 'biosensors' for chemicals of concern at their point of entry into ecosystems. These tools could be invaluable for the next generation of environmental monitoring and pollution control.

## Methods

### Bacterial isolation and culture
We tested 12 strains of bacteria in this experiment (Supplementary Table 2). Ten of these were environmental isolates from Iceland (previous papers give more detailed site descriptions[35,36]). These isolates were cultured from sediment samples obtained from pristine freshwater streams (Supplementary Fig. 8), that is, from a landscape free from agriculture or urbanization (and therefore likely to have no history of chemical exposure). Sediment samples from Icelandic streams were collected in 30% $v/v$ glycerol (final concentration; Sigma-Aldrich) and stored at −80 °C until required. Bacterial isolates were picked from colonies grown on 1/10 R2A and R2A agar plates (Sigma-Aldrich) and stored at −80 °C in ProtectTube Cryobead Systems (Technical Consultants). DNA was extracted using the ZR-96 Fungal/Bacterial DNA extraction kit (Zymo Research). The 16S ribosomal RNA gene was amplified for sequencing using the 27F (5′-AGAGTTTGATCCTGGCTCAG-3′) and 1492R (5′-TACGGYTACCTTGTTACGACTT-3′) primer pair[37], and taxonomy of isolates was determined via BLAST (v2.15.0).

We selected a range of strains from this isolate library spanning a broad phylogenetic diversity (Supplementary Table 2). Two of the species (*P. baetica* and *A. humicola*) were captured twice, at different locations, allowing us to investigate the consistency of stressor impacts (Supplementary Fig. 8).

We additionally tested two strains of lab bacteria: *A. fischeri* and *E. coli* K-12. *A. fischeri* is the active agent of the Microtox assay widely used in ecotoxicology[15]. *E. coli* has also been frequently used in chemical toxicity assays[16]. These were selected as a comparison to the Iceland bacteria, to understand whether the responses of lab bacteria widely used in toxicity testing can be generalized to bacteria from pristine systems.

All bacteria were revived from frozen stocks and grown to carrying capacity in Luria–Bertani (LB) media (Sigma-Aldrich) before chemical toxicity testing. *A. fischeri* is a marine bacterium and will only grow in high-salinity media; the LB media for this strain was therefore supplemented with sodium chloride to 20 g l⁻¹ (ref. 38).

### Chemical treatments
We built stressor mixtures from eight chemicals representing a range of classes of pollutants which are known for their prevalence in freshwater environments[34] (Supplementary Table 3). These pollutants represent four major groups of stressors targeting different components of freshwater ecosystems and have been identified as key targets for microbial studies[34]. As the effects of pollutants on non-target groups are often overlooked in freshwater ecology, bacterial EC$_{50}$ (half maximal effective concentration) data are generally not available for the non-antibiotic chemicals used here. Based on preliminary work, we chose a concentration of 0.1 mg l⁻¹, a dose that elicits a response in at least one strain of bacteria for each chemical tested (Supplementary Fig. 1). For each chemical, this dose is also within an order of magnitude of the effective concentration for at least one non-bacterial taxa group, based on the US Environmental Protection Agency EcoTox database[39], with the exception of diflufenican (EC$_{50}$ 0.001–0.008 mg l⁻¹ in algae), that is, a concentration of realistic concern to other parts of the ecosystem (Supplementary Table 4). To most realistically represent chemical pollutants entering the environment, where possible we used pesticide products containing these stressors as their active ingredients, rather than purified versions of the chemicals (Supplementary Table 3). This was not feasible for metaldehyde, which is generally supplied as insoluble slug pellets, so here we used the purified chemical form (Supplementary Table 3).

### Multiple chemical stressor experiments
For each bacterial strain, microcosms were set up in 96-well plates with each possible combination of chemical stressor (all at 0.1 mg l⁻¹) diluted in LB (supplemented with sodium chloride to 20 g l⁻¹ for *A. fischeri*).

Bacteria were added such that the final mixtures contained a 1 in 100 dilution of bacterial culture from carrying capacity. We also created a mixed culture of the pristine strains from Iceland. Here we mixed equal volumes of the 10 strains of bacteria at carrying capacity and added this culture to the chemicals in LB such that the final mixtures contained a 1 in 100 dilution of the whole mixed culture. Therefore, in the mixed culture experiments each of the 10 strains will be at 1/10 its density in monoculture; however, the whole mixture will contain approximately the same overall quantity of bacteria as the monoculture experiments.

Growth was assayed by measuring the absorbance of the cultures at 600 nm ($A_{600}$) once per hour, for 72 h. Plates were briefly shaken before reading to homogenize samples and disrupt biofilm formation. The full set of combinations of 8 chemicals produces 255 possible mixtures, to which we added 45 controls of bacteria in fresh media without chemical addition. These microcosms were set up in replicates of 4 to produce 1,200 total growth curves per bacterial isolate.

## Quantifying multi-chemical stressor interactions

We used the AUC as a fitness metric[40,41]. Bacterial growth curves offer many other aspects to study, such as lag time (length of the lag phase before exponential growth begins), maximum growth rate and carrying capacity[42]. However, picking a single focal parameter was not appropriate, as a stressor may affect any or all of them, so by using AUC we combined all growth phases into a single parameter which is correlated positively with both the growth rate and the carrying capacity[40]. We fitted a spline function to each growth curve and integrated across it over a fixed time period (72 h) to calculate the AUC.

We tested the effects of stressors as the ratio of the AUC of growth in the presence of stressor(s) versus that in control conditions (no stressors), yielding a measure of relative growth:

$$G_i = AUC_i/AUC_{control}$$

where $i$ stands for a specific stressor or a combination of stressors. Similar to the approach taken by Tekin et al.[43], we calculated two measures of the structure of interactive effects: net interactions among stressor in a mixture (not disentangling specific interactions between stressors) and emergent interactions (specific higher-order interactions between stressors).

The net interaction measure simply considers the bacterial growth in mixture compared to a multiplicative null model containing terms for the responses to individual chemicals only. We considered multiplicative, and not additive, null models because we measured relative fitness, which is equivalent to a percentage change in growth. Using an additive model for relative fitness could lead to biologically meaningless predictions. For example, if two stressors each reduce bacterial growth by 70%, then an additive model would predict their combination to reduce growth by 140%. This is not meaningful as a reduction in growth must be bounded at 100%, that is, zero measurable growth. Using a multiplicative model, we would predict a 91% reduction in growth in this example ($1 - (1 - 0.7)^2 = 0.91$). Therefore, the null model for the combined effects must be the product of the percentages, not the sum[19].

For example, for a mixture of three chemicals X, Y and Z, we measured relative growth ($G$) in the presence of each stressor ($G_X$, $G_Y$ and $G_Z$, respectively) and in the presence of all three combined ($G_{XYZ}$), calculated using the above measure. The null model for the net effect is simply the product of relative growth in the presence of each stressor:

$$G_X G_Y G_Z$$

The net interaction term ($N_A$), where $A$ refers to a particular combination (set) of stressors, can be calculated in our example here as follows:

$$N_{XYZ} = \frac{G_{XYZ}}{G_X G_Y G_Z} \tag{1}$$

If $N_A$ is significant (based on bootstrapping; see 'Testing interaction significance'), this tells us that growth in this mixture deviates from the null model; that is, there is a net interaction between the stressors (Supplementary Fig. 9a). This net interaction ($N_{XYZ}$) may be due to a significant higher-order 'emergent' (in this case just one possible three-way) interaction, in addition to the two-way (pair-wise) interactions among the three stressors. To test for the presence of a significant emergent interaction, we need to account for all such lower-order interactions within the mixture, which requires measurement of relative growth under each lower-order stressor combination as well[43]. Thus, for the three-way example, the growth in mixture accounting for all possible two-way interactions is given by the following:

$$G_{XYZ} = G_X G_Y G_Z I_{XY} I_{XZ} I_{YZ} I_{XYZ},$$

where $I_{XY}$, $I_{XZ}$ and $I_{YZ}$ are the two-way interactions and $I_{XYZ}$ is the three-way emergent interaction (Supplementary Fig. 9b). We can therefore calculate the emergent interaction as follows:

$$I_{XYZ} = \frac{G_{XYZ}}{G_X G_Y G_Z I_{XY} I_{XZ} I_{YZ}}, \tag{2}$$

given data on all the two-way interactions.

For any given set of stressors, $A$, we can test for an emergent interaction, $I_A$, provided that we have observations of growth $G_K$ under all unique combinations of stressors from $A$. To frame this formally, we use $P(A)$ to indicate the power set of $A$, that is, the set of all its possible subsets.

First, we define $K \subset P(A)$ by $K = \{L \in P(A), 2 \le |L| < |A|\}$, that is, the set of all unique subsets of $A$ meeting the condition of containing at least two elements (stressors) but not containing all the stressors from $A$. We then iterate through these subsets in the order of the number of stressors they contain to calculate the following:

$$I_A = \frac{G_A}{(\Pi_{a \epsilon A} G_a)(\Pi_{k \epsilon K} I_k)} \tag{3}$$

where the first product term in the denominator represents growth under each stressor individually, and the second product term represents interaction terms with combinations of stressors excluding the case of all stressors in $A$ combined. To calculate this in practice is an iterative process starting with calculation of interaction terms for pairs of stressors and then sets of three and so on. At each level of increasing stressor mixture complexity, we incorporate all the previously calculated interaction coefficients into our calculations.

## Testing interaction significance

We used bootstrapping to test the significance of chemical responses and interaction effect sizes. First, we tested whether the chemical mixture had a significant impact on growth. For each chemical mixture, we resampled the experimental data with replacement (that is, control growth and growth in the presence of the chemical mixture) and calculated $G_i$ as above. We repeated this 10,000 times to generate a distribution of $G_i$ and calculated the 95% confidence intervals. If the confidence interval included 1, we interpreted these as showing no response to the chemical mixture. For mixtures with confidence intervals excluding 1 (that is, those showing a significant response to the chemical mixture), we proceeded to test the significance of net and emergent interaction effects following the same bootstrapping procedure. For each interaction test, we resampled the relevant replicated experimental data (that is, data points corresponding to each term within equations (1) or (3)) with replacements and calculated the interactions according to equations (1) and (3). We repeated this 10,000 times to generate distributions of net and emergent interaction terms and calculated the 95% confidence intervals. If the 95% confidence

intervals excluded 1 (which would correspond to no interaction), we interpreted these as interaction effects that deviated significantly from the null (multiplicative) model expectations.

Once the significance of interactions was tested, we determined the type of interaction (synergism or antagonism). Detecting interactions among ecological stressors can be problematic when stressors can elicit positive or negative responses depending on the context[44]. For example, bacterial taxa may metabolize and grow using chemicals that are toxic to others[45], including the catabolism of antibiotics[46]. Defining interactions as antagonistic or synergistic is further complicated when individual stressors operate in opposing directions, as seems to be commonplace[11,30,44]. Previous work to detect high-order interactions in microbial responses to stressors has generally not accounted for stressors operating in opposing directions, focusing on negative effects only[13,19]. Here we extend previous work to account for both positive and negative effects. To allow for both positive and negative effects of stressors on growth, we defined significant interactions as antagonistic or synergistic as follows[30,47]. We first asked in what direction the null expectation was (reduction versus increase in growth) and used this as the basis for defining interactions. If the null expectation was a positive effect (increase in growth), then interaction terms >1 were synergistic (growth in the mixture is more positive than predicted), while interaction terms <1 were antagonistic (growth in mixture is less positive than predicted). Conversely, if the null was a reduction in growth, then interaction terms <1 were synergistic and interaction terms >1 were antagonistic (see Supplementary Fig. 9).

All analyses were performed in R version 4.2.1 (ref. 48).

### Testing for phylogenetic constraints on bacterial responses

We used 16S sequences to construct a phylogeny of the bacterial strains used in this experiment. For the Iceland strains, we used the 16S sequences directly collected from those strains; for *E. coli* and *A. fischeri*, we obtained reference sequences from National Center for Biotechnology Information GenBank (*E. coli* accession, MW349588.1; *A. fischeri* accession, FJ464360.1). Sequences were aligned in MAFFT (v7.205), and from this alignment a phylogeny was inferred in RAxML (v8.1.1) using a general time reversible substitution model with gamma distributed rates.

For each bacterial strain, we used the relative growth, $G_A$, in every chemical stressor combination as an overall phenotype for stressor response. We calculated the pair-wise Euclidean distance of stressor responses between strains and then visualized the similarity between strains using hierarchical clustering.

We extracted the pair-wise distance matrices for both the phenotypic data (the clustered chemical responses) and the phylogeny and tested the association between them using a Mantel test. This test is appropriate when measuring phylogenetic signal from multiple continuous traits, that is, our clustering based on phenotypic responses to 255 chemical mixtures[49]. Here we used Kendall's rank correlation $\tau$ as the statistical method applied to the Mantel test due to the non-parametric distribution of the pair-wise phenotypic distances.

To further test evolutionary constraints on the responses of bacteria to each of the chemicals tested here, we quantified two measures of phylogenetic signal—Pagel's $\lambda$ (ref. 50) and Blomberg's $K$ (ref. 51). These metrics both quantify the degree to which the distribution of traits at the tips of a phylogeny (here given by our $G$ metric) are driven by shared evolutionary history. For both metrics, values of 0 imply no phylogenetic signal, whereas values of 1 imply strong phylogenetic signal, where the trait has evolved gradually along the phylogenetic tree under an evolutionary model approximating Brownian motion. $\lambda$ is bounded between 0 and 1, with intermediate values showing deviation from a Brownian motion model due to factors such as variation in evolutionary rate over time. $K$ is not bounded in such a way, and $K > 1$ indicates more phylogenetic signal than expected under a Brownian motion model, implying a substantial degree of trait conservatism.

Pagel's $\lambda$ requires that the trait be normally distributed; however, our $G$ values tended to have left-skewed distributions. Therefore, we transformed the values to produce a normal distribution before testing for $\lambda$.

The Mantel test and test for significance were performed using the Vegan R package, version 2.6-4 (ref. 52). The $\lambda$ and $K$ tests for phylogenetic signal were performed with the Phytools R package, version 1.2-0 (ref. 53).

### Statistics and reproducibility

No statistical method was used to predetermine the sample size. If there was evidence of contamination in any of the bacterial growth curves, these data were excluded from the analyses. Then, 383 of 15,120 curves were excluded due to potential contamination (2.5%). The experiments were not randomized. The Investigators were not blinded to allocation during experiments and outcome assessment.

### Reporting summary

Further information on research design is available in the Nature Portfolio Reporting Summary linked to this article.

### Data availability

The minimum dataset necessary to interpret, verify and extend the research is available to download at https://doi.org/10.6084/m9.figshare.25054913.v1. All data are also available in our GitHub repository at https://github.com/smithtp/isolate-chem-mixtures/. The 16S sequences for the isolates used have been deposited in the NCBI GenBank under accession codes PP204206:PP204215. The US Environmental Protection Agency EcoTox database accessed for effective concentrations is available at https://cfpub.epa.gov/ecotox/.

### Code availability

Documented code to replicate all analyses in this manuscript is available from our GitHub repository at https://github.com/smithtp/isolate-chem-mixtures/

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

## Acknowledgements

We thank M. Beeby (Imperial College London, UK) for supplying us with a culture of *Aliivibrio fischeri*. All authors were funded by a Natural Environment Research Council grant NE/S000348/1.

## Author contributions

T.P.S., T.B. and E.R. designed the experiments. T.P.S. and T.M.-L. performed the experiments. T.C., T.P.S., S.P. and J.R. devised the interactions analysis. T.P.S. collated the data and performed the analyses. All authors interpreted the results. T.P.S. drafted the manuscript. All authors contributed to revising the manuscript. G.W., T.B., S.P., J.R. and E.R. acquired the funding.

## Competing interests

The authors declare no competing interests.

## Additional information

**Correspondence and requests for materials** should be addressed to Thomas P. Smith.

# Reporting Summary

## Statistics

For all statistical analyses, confirm that the following items are present in the figure legend, table legend, main text, or Methods section.

| n/a | Confirmed | |
|---|---|---|
| ☐ | ☒ | The exact sample size (*n*) for each experimental group/condition, given as a discrete number and unit of measurement |
| ☐ | ☒ | A statement on whether measurements were taken from distinct samples or whether the same sample was measured repeatedly |
| ☐ | ☒ | The statistical test(s) used AND whether they are one- or two-sided<br>*Only common tests should be described solely by name; describe more complex techniques in the Methods section.* |
| ☒ | ☐ | A description of all covariates tested |
| ☐ | ☒ | A description of any assumptions or corrections, such as tests of normality and adjustment for multiple comparisons |
| ☐ | ☒ | A full description of the statistical parameters including central tendency (e.g. means) or other basic estimates (e.g. regression coefficient) AND variation (e.g. standard deviation) or associated estimates of uncertainty (e.g. confidence intervals) |
| ☐ | ☒ | For null hypothesis testing, the test statistic (e.g. *F*, *t*, *r*) with confidence intervals, effect sizes, degrees of freedom and *P* value noted<br>*Give P values as exact values whenever suitable.* |
| ☒ | ☐ | For Bayesian analysis, information on the choice of priors and Markov chain Monte Carlo settings |
| ☒ | ☐ | For hierarchical and complex designs, identification of the appropriate level for tests and full reporting of outcomes |
| ☒ | ☐ | Estimates of effect sizes (e.g. Cohen's *d*, Pearson's *r*), indicating how they were calculated |

*Our web collection on statistics for biologists contains articles on many of the points above.*

## Software and code

Policy information about availability of computer code

| Data collection | No software was used for data collection. |
|---|---|
| Data analysis | Data analysis packages:<br>MAFFT (v7.205); RAxML (v8.1.1); R (version 4.2.1); Vegan R-package (version 2.6-4); Phytools R-package (version 1.2-0).<br>All custom R code for our analyses are available from: https://github.com/smithtp/isolate-chem-mixtures/ |

For manuscripts utilizing custom algorithms or software that are central to the research but not yet described in published literature, software must be made available to editors and reviewers. We strongly encourage code deposition in a community repository (e.g. GitHub). See the Nature Portfolio guidelines for submitting code & software for further information.

## Data

Policy information about availability of data

All manuscripts must include a data availability statement. This statement should provide the following information, where applicable:
- Accession codes, unique identifiers, or web links for publicly available datasets
- A description of any restrictions on data availability
- For clinical datasets or third party data, please ensure that the statement adheres to our policy

The minimum dataset necessary to interpret, verify and extend the research is available to download from figshare: https://doi.org/10.6084/m9.figshare.25054913.v1

All data is also available in our GitHub repository: https://github.com/smithtp/isolate-chem-mixtures/
16S sequences for the isolates used have been deposited in NCBI GenBank under accession codes PP204206:PP204215.
The EPA EcoTox database accessed for effective concentrations is available at: https://cfpub.epa.gov/ecotox/

## Research involving human participants, their data, or biological material

Policy information about studies with human participants or human data. See also policy information about sex, gender (identity/presentation), and sexual orientation and race, ethnicity and racism.

| | |
|---|---|
| Reporting on sex and gender | N/A |
| Reporting on race, ethnicity, or other socially relevant groupings | N/A |
| Population characteristics | N/A |
| Recruitment | N/A |
| Ethics oversight | N/A |

Note that full information on the approval of the study protocol must also be provided in the manuscript.

## Field-specific reporting

Please select the one below that is the best fit for your research. If you are not sure, read the appropriate sections before making your selection.

☒ Life sciences          ☐ Behavioural & social sciences          ☐ Ecological, evolutionary & environmental sciences

For a reference copy of the document with all sections, see nature.com/documents/nr-reporting-summary-flat.pdf

## Life sciences study design

All studies must disclose on these points even when the disclosure is negative.

| | |
|---|---|
| Sample size | Sample sizes were not statistically predetermined. he general standards of the field are to provide at least 3 independent biological replicates for each condition (see e.g. https://doi.org/10.1002/bies.202100108). We struck a balance between maximizing statistical power whilst still leveraging the high-throughput nature of our experiments, by performing 4 independent biological replicates for each condition. |
| Data exclusions | Growth curves in negative control wells were inspected by eye for evidence of contamination. Where contamination was observed, data from the plate was excluded. 2.5% of growth curves were excluded. |
| Replication | Experimental growth assays were performed in 4 independent biological replicates. Where data was excluded (see exclusions), at least 3 biological replicates from each condition were taken forward for analysis. |
| Randomization | Randomization was not relevant as we were applying a uniform set of treatments to each strain of bacteria. |
| Blinding | Blinding was not relevant as each strain of bacteria had the same set of treatments applied to it, and data was then collected continuously by a robotic plate reader. |

## Reporting for specific materials, systems and methods

We require information from authors about some types of materials, experimental systems and methods used in many studies. Here, indicate whether each material, system or method listed is relevant to your study. If you are not sure if a list item applies to your research, read the appropriate section before selecting a response.

### Materials & experimental systems

| n/a | Involved in the study |
|---|---|
| ☒ | Antibodies |
| ☒ | Eukaryotic cell lines |
| ☒ | Palaeontology and archaeology |
| ☒ | Animals and other organisms |
| ☒ | Clinical data |
| ☒ | Dual use research of concern |
| ☒ | Plants |

### Methods

| n/a | Involved in the study |
|---|---|
| ☒ | ChIP-seq |
| ☒ | Flow cytometry |
| ☒ | MRI-based neuroimaging |

