## [Peer Review File · Nature Microbiology]

Peer Review Information

Journal: Nature Microbiology

Manuscript Title: High throughput characterization of bacterial responses to complex mixtures of chemical pollutants

Corresponding author name(s): Dr Thomas Smith

Reviewer Comments & Decisions:

Decision Letter, initial version:

Message: 9th May 2023

Dear Dr Smith,

First of all, I apologize again for the delay in getting back to you, and I thank you very much for your patience while your manuscript "Bacterial responses to complex mixtures of chemical pollutants" was under peer-review at Nature Microbiology. It has now been seen by 3 referees, whose expertise and comments you will find at the end of this email. Although they find your work of some potential interest, they have raised a number of concerns that will need to be addressed. We are very interested in moving forward with your work at Nature Microbiology, but in light of their comments we will need to see a revised manuscript before we can consider publication of the work.

In particular, the overarching concerns of the referees center around a request for additional analyses and discussion with the goal of synthesizing a more striking conclusion that highlights the power of your complex experimental approach. Should further experimental data allow you to address these criticisms, we would be happy to look at a revised manuscript. I would also be happy to discuss a revision plan with you before you set to work on the revisions, so as to ensure we are all on the same page. Just let me know!

Please include a data availability statement as a separate section after Methods but before references, under the heading "Data Availability". This section should inform readers about

2the availability of the data used to support the conclusions of your study. This information includes accession codes to public repositories (data banks for protein, DNA or RNA sequences, microarray, proteomics data etc...), references to source data published alongside the paper, unique identifiers such as URLs to data repository entries, or data set DOIs, and any other statement about data availability. At a minimum, you should include the following statement: "The data that support the findings of this study are available from the corresponding author upon request", mentioning any restrictions on availability. If DOIs are provided, we also strongly encourage including these in the Reference list (authors, title, publisher (repository name), identifier, year). For more guidance on how to write this section please see:
<http://www.nature.com/authors/policies/data/data-availability-statements-data-citations.pdf>

* If you have not done so already we suggest that you begin to revise your manuscript so that it conforms to our Article format instructions at <http://www.nature.com/nmicrobiol/info/final-submission>. Refer also to any guidelines provided in this letter.

When submitting the revised version of your manuscript, please pay close attention to our [href="https://www.nature.com/nature-portfolio/editorial-policies/image-integrity">Digital Image Integrity Guidelines. and to the following points below:](https://www.nature.com/nature-portfolio/editorial-policies/image-integrity)

Note: This url links to your confidential homepage and associated information about manuscripts you may have submitted or be reviewing for us. If you wish to forward this e-mail to co-authors, please delete this link to your homepage first.

Nature Microbiology is committed to improving transparency in authorship. As part of our efforts in this direction, we are now requesting that all authors identified as 'corresponding author' on published papers create and link their Open Researcher and Contributor Identifier (ORCID) with their account on the Manuscript Tracking System (MTS), prior to acceptance. This applies to primary research papers only. ORCID helps the scientific community achieve unambiguous attribution of all scholarly contributions. You can create and link your ORCID from the home page of the MTS by clicking on 'Modify my Springer Nature account'. For more information please visit www.springernature.com/orcid.

If you wish to submit a suitably revised manuscript we would hope to receive it within 6 months. If you cannot send it within this time, please let us know. We will be happy to consider your revision, even if a similar study has been accepted for publication at Nature Microbiology or published elsewhere (up to a maximum of 6 months).

Reviewer Expertise:

Referee #1: microbial ecology, ecotoxicology, chemical stressors

Referee #2: environmental microbiology, ecotoxicology, applied microbiology

Referee #3: microbial ecology

Reviewer Comments:

Reviewer #1 (Remarks to the Author):

Smith et al present an interesting study on the response of diverse bacterial strains to stressor mixtures. This work is truly novel and timely in bacteriology and ecotoxicology that rarely consider multiple stressors (at least more than 2) or diverse environmental strains. The authors present an important body of work with a large scale experiment and in-depth statistical analyses. The conclusions of the paper on the interactions between stressors are novel and not presented in other bacterial studies to my knowledge. My main comments are that I feel that some important figures or analyses have been left out of the manuscript to strictly focus on interactive effects, while I think that describing the absence of effects or additive effects are also important to report. The authors also keep their results and discussion at a very broad level, not describing the specific combinations of stressors that generate significant interactions. I think that for instance reporting that some stressors or types of stressors (herbicide, antibiotic...) tend to interact

3(synergistically or antagonistically) more frequently than others are important results for the scientific community.

Specific comments:

Figure 1: Indicate in the legend or the text that a value of G below 1 corresponds to a decrease in growth. You could add a line at 1 and indicate that values above have increased growth while values below have decreased growth

Figure 2: resolution of panels B and C are not great and have low legibility

Panel B: could you add colors/ info on the phylogeny of the different strains. The readers start reading the paper with the results first and don't even know the genera of the strains studied (shortened just to the first letter) and if they belong to diverse phyla or bacterial families

Panel C: The points could be colored based on the strains and the color coding could be nuanced based on the bacterial families (ex: all Pseudomonadaceae have different shades of green).

Panel A: Similarly, when we start reading the results associated with Figure 2, no explanation has been provided in the text on the type of stressors selected and their main "categories" (antibiotic, herbicide...) or the concentrations used. Maybe provide 1-2 sentences on the rationale in the result section.

Quantifying multi-stressor interactions:

The approach proposed to assess stressor interactions is interesting and well explained. I just wonder why the authors only focused on interactive effects and didn't present also the results on the number of combinations that presented additive effects or no effects at all. These results are also interesting to determine the prevalence of no effect vs additivity vs antagonism/synergism and inform on the prevalence of additive effects in stressor mixture to better predict environmental effects (Additivity is not as bad as synergism but still it is not great).

The panel A of figure 3 suggests that the "norm" or null hypothesis is that these stressors decrease bacterial growth and that stressors have additive effects but I wonder how often it is verified. Some stressors will have no effects at all or will increase growth. Here you display the results just based on the presence of interactive effects or not, but no info is presented on the directionality (positive/negative effects) or the proportion of treatments having no effects at all or the number of treatments presenting no interactions due to additivity (currently all presented under the "no interaction" category).

Maybe you could present or explore these other categories of responses and show some "real life" examples to describe the diversity of responses observed.

For Figure 2 and Figure 3, I wonder if the analyses should not be done in presence and absence of Oxytetracycline like you did in Figure 1. We could imagine that the number of interactions might increase because of the "dominant" effect of Oxytetracycline basically killing most bacteria might hide interesting patterns between other stressors.

I have the feeling that some important figures have not been presented in the main text or

4SI. Only 5 figures are presented in the main text and 3 in SI, this looks to me that some important aspects of the datasets have not been exploited. The heatmap in figure 2 is a nice visualization but do not provide statistical tests for some conclusions related to the effect of the number of stressors (L121-122) or the similarity of responses between different strains. Simple figures showing the relationship between G and the number of stressors for each strain (each strain is a different line) or the prevalence of interactive effects for each stressor or some interesting stressor combinations could be a nice addition. In the end, we don't really have a clear idea of the "interesting" or "surprising" interactions observed, whereas they are antagonistic or synergistic. The manuscript stays descriptive at a high level and do not provide specific examples that could provide important new knowledge in ecotoxicology. Which stressors or types of stressors interact more frequently? In this case, which associations generate antagonistic or synergistic interactions? In the discussion, you could add a section to discuss potential mechanisms associated with the interaction observed and illustrate it with some selected examples. How antagonism or synergism can emerge between an antibiotic and a herbicide?

Methods:

Choice of concentration at 0.1 mg/L: I understand the rationale but why not have used specific concentrations for each stressor based on either their environmental concentrations or dose response curves (EC50 or other)? Of course, the experiment cannot be redone but can you amend the table 3 with this information when they are available?

L352: I'm not sure to understand how you obtain the number of 255 possible mixtures for the full set of combinations of 8 stressors. I would have assumed that this number will be 8 factorial $7 = 40320$. Of course, this is not feasible in a lab experiment but can you clarify the design?

Reviewer #2 (Remarks to the Author):

This manuscript on "Bacterial responses to complex mixtures of chemical pollutants" intends to address a very important and significant issue on the general topic of applied toxicology or microbial toxicology, but both the experimental design and results do not allow to advance the topic to a higher level or a significant extent in my understanding of this subject. Because of this, I feel that more specific focuses shall be given on the selection of methods, effective concentration for toxicity, mechanisms involved, the biochemical and physiological responses of the microorganisms so that the subject can benefit from the results for further advancement.

1) Microbial responses can be assessed by a arrange of determinants, but biomass by optical density is not sensitive enough because both the live and dead cells in the population were accounted without any differentiation of them, so the value is a mixture of cells of different physiological stages and also the dead ones.

A better and more sensitive measurement of the cells can be identified and used to achieve a more sensitive response from the metabolically active microorganisms. On the general side, heat-shot proteins and responsible genes can be used while biochemical reaction

5

related genes can be selected for the others. In this way, the toxicity can be evaluated to exclude the metabolism involved.

2) Chemicals and concentrations: Toxicology addresses the issue of chemical concentration effects on any selected organisms, but the concentration expressed and used in experiments and report shall be the active or effective concentrations, not simply the total mass per volume as targeted ones. This is very critical and has been very important to the pharmacology in medicine in evaluation of the chemical dosage to cure or kill the microorganisms for different age groups. Unfortunately, applied toxicology, ecotoxicology and environmental toxicology have been using the concentration of the environment and environmental samples too loosely by the individual involved. This is particularly true in applied and ecotoxicology when a given amount of chemicals are introduced into a testing system of marine or freshwater without checking of the soluble or effective concentration of the target chemical, and the results are registered for the supposedly chemicals for correlation with the toxic effects detected. This practice is not only wrong for the sake of science, this has generated a large data set with very low reproducibility and very high variability. Such situation prevents science for a healthy development to move to a higher level and also laying a solid foundation for future development.

The selection of the 8 chemicals in this investigation did not seem to have any intended plan for the different chemical structures and their effects on the selected microorganisms. The justification of their widely occurrence is sound to some extent, but the results from such understanding cannot advance the in-depth science further to enhance the research subject to grow.

For any chemical pollutant, there are potentially concentration of bacterial maintenance, metabolism, and then toxicity. This framework can be used to lay a good foundation for the toxicity testing and analysis. Prevalence of them is one way to justify for their relevance and significance. The chemical structures of them shall be carefully selected to yield useful information on the possible mechanism of toxicity or the gene expression so that the selective markers for monitoring can be chosen more specifically.

The concentration of 0.1 mg/L is apparently far too high for many of the 8 chemicals to be soluble in water. Because of this, the toxicity contribution from each of the chemicals is not on an even shared basis or truly concentration determined phenomena. Furthermore, interactions of different chemicals also may result in insolubility and different effects from common exposure, additive, antagonism, mutualism and also potentiation.

3) The organisms of choice: Commonly used ones are included to a large part, but *Salmonella* sp. of the Ames test for mutagenesis was not included, which is a pity because a very large database is available based on the Ames test and this study can benefit from the data and make comparison with the data available.

The phylogeny of the selected bacteria was used for possible logical information with the toxic chemicals. This part is not likely to yield anything useful for at least two reasons. First, 16S rRNA gene is used for taxonomy and classification mostly and this gene does not contain fragments for metabolisms or adaptation. In addition, genomic DNA does not yield

the activity-based information of the microorganisms, so RNA shall be used to closely monitor the metabolic activity microorganisms and gene expression.

4) The ecology approach using fitness concept and model does not advance well on the subject matter because the competition for survival and growth is not based on resource available in this case, but the toxic chemicals on their survival and metabolism. Assuming some chemicals may be utilized by some microorganisms, then the others will need to deal with the toxic effect for survival. But the selection of the chemicals was not carefully delineated initially for the experiments, so there was not logic or plan for this part to delineate the information from the well-planned experiments.

The results in Fig. 1 are predictable for the trend and it is not new.

The chemical stressors and responses in Fig. 2 are simply presenting the data without any synthesis or conclusion can be derived from them. Such a fact reflects the serious weakness in the initial experimental design and conceptual model.

In addition, the multi-stressors and interactions in Fig. 3 do not reveal anything of specific information and mechanism to allow a better understanding of this topic.

The interaction analysis shown in Fig. 4 is not based on a well analyzed background of these chemicals with their effective concentration so that the effects or connectivity can be interpreted.

The interactions of net or emergent in Fig. 5 require the concentrations of these chemicals and their toxicity must be known before making any claim of the relationship.

Reviewer #3 (Remarks to the Author):

Broader comments

The authors conducted an ecotoxicology study on the effect of 8 stressors on the growth of 12 bacterial strains. They tried to fill two research gaps: 1) lack of studies with more than 3 stressors especially for the interactions among them and 2) most studies are based on a few model organisms. Their analytical approaches to quantify net and emergent interactions on the growth of bacterial strains are generally convincing yet some aspects and results may raise concern (see below for the specific comments). Also, the interactions at the other side of the equation should have not been ignored as well by including the mix-culture of all 12 strains as additional culture (even though it was mentioned for the future direction)

Specific comments

L129-131: Although the Mantel test is a straightforward way to measure two dissimilarity matrices, it often is abused by overly extending from its original intent of testing dissimilarity per geographic distance. Figure 2b shows some interesting patterns that may not be well revealed by the Mantel test. There are approaches to compare two tree topologies such as comparePhylo function from ape package. The p value of 0.156 is not

7

exceptionally high to fail to reject the null hypothesis, and I would like to see some more detailed analysis and discussions from figure 2b.

Figure 1: Linear regression results for with and without oxytetracycline in the legend are incomplete. Please include all the test statistics for both with and without oxytetracycline.

L146-147 & Figure 3d & e: The conceptual distinction between net interactions and emergent interactions is clear, but I feel that there may be an alternative approach to measure emergent interactions. Figure 3e is counter-intuitive in that emergent interactions decrease as the number of stressors increases. I feel this may be an artifact from the current implementation to account for all possible subsets of chemical stressors for emergent interactions including lower-order emergent interactions, which may underestimate the emergent interactions by overestimating lower-order (emergent) interactions.

L213: Yes, but the important step in between should be on the response as a community of microbes for the biological interactions, since they have very important role in ecosystem processes.

L227-229: Using "similar" prevalence of antagonistic and synergistic responses as evidence to use microbes for macro-organisms reminds me of the old ecological research trend of creating microbial "black box" since it doesn't account for the apparent distinction of physiology between them. However, I can agree with the authors that microbes can be great bioindicators for ecotoxicological studies, but the reason is because of the significant role of microbes in ecosystem stability and function.

L231-233: Another good reason the biotic interactions among bacterial strains should be considered for a more comprehensive understanding here.

L241-243: I'd like to see more reasoning on how insignificant associations between phylogenetic and phenotypic distribution support convergent evolution on physiological responses.

L252-271: I appreciate the proposed future work focusing on microbial communities. However, or because of this, I can't really agree with the implications the authors try to make based on the relationship between species richness and multifunctionality. I think it is an overly stretched point and removing it may make the latter half of the paragraph more convincing.

L377-381: The use of a multiplicative null model need more explanation. It may not be just me having a hard time being convinced why the combined effects based on relative fitness measures must be the products, not the sum. Cited work by Tekin et al. 2018 in npj doesn't seem to support the use of the multiplicative model, as their null model of multiway interactions is based on an additive model.

L442-450: Does any of the null expectation was positive with those stressors? I think including positive null expectation is unnecessary and may cause confusion to some readers. Both figures 3a and 5 are based on negative null expectations only.

L468: Do this phenotypic data refer to the growth response to stressors?

L474: Report the version of vegan package as you did for R on line 452.

Author Rebuttal to Initial comments

Reviewer Expertise:

Referee #1: microbial ecology, ecotoxicology, chemical stressors

Referee #2: environmental microbiology, ecotoxicology, applied microbiology

Referee #3: microbial ecology

Reviewer Comments:

Reviewer #1 (Remarks to the Author):

1. Smith et al present an interesting study on the response of diverse bacterial strains to stressor mixtures. This work is truly novel and timely in bacteriology and ecotoxicology that rarely consider multiple stressors (at least more than 2) or diverse environmental strains. The authors present an important body of work with a large scale experiment and in-depth statistical analyses. The conclusions of the paper on the interactions between stressors are novel and not presented in other bacterial studies to my knowledge

My main comments are that I feel that some important figures or analyses have been left out of the manuscript to strictly focus on interactive effects, while I think that describing the absence of effects or additive effects are also important to report. The authors also keep their results and discussion at a very broad level, not describing the specific combinations of stressors that generate significant interactions. I think that for instance reporting that some stressors or types of stressors (herbicide, antibiotic...) tend to interact (synergistically or antagonistically) more frequently than others are important results for the scientific community.

We thank the reviewer for their constructive comments. We had attempted to keep the discussion and conclusions broad so that we could derive generalisations about the importance of interactions in microbial ecotoxicology more broadly, rather than focusing on specific interactions uncovered within our work. Rather than discussing each of the many chemical interactions that we observed, we now delve more deeply into selected stressor combinations and potential interaction mechanisms in our discussion (see our responses to specific comments below).

In considering our response to this comment, we have substantially modified several figures, presented additional figures and modified the analyses to include the absence of effects. Responses to individual comments are provided below.

Specific comments:

2. Figure 1: Indicate in the legend or the text that a value of G below 1 corresponds to a decrease in growth. You could add a line at 1 and indicate that values above have increased growth while values below have decreased growth

We have made this change, the revised figure (now figure 2) has a line added indicating values with increased and decreased growth.

3. Figure 2: resolution of panels B and C are not great and have low legibility

We have revised this figure and now present the phylogenetic results in a separate, more legible figure (Fig 3) based on both this comment, and in response to Reviewer #3's comment #2 on phylogenetic testing. The previous panel C (distance matrix comparison) has been moved to a separate figure in the supplement (Supplementary Figure S3).

Issues with resolution may be a function of MS Word not properly importing figures at high resolution. High resolution versions of all figures will be provided to the journal.

104. Panel B: could you add colors/ info on the phylogeny of the different strains. The readers start reading the paper with the results first and don't even know the genera of the strains studied (shortened just to the first letter) and if they belong to diverse phyla or bacterial families

Panel C: The points could be colored based on the strains and the color coding could be nuanced based on the bacterial families (ex: all Pseudomonadaceae have different shades of green).

We have addressed this by adding colour-coded taxonomic information to the phylogeny figure to highlight the phylogenetic diversity present (now Figure 3a).

5. Panel A: Similarly, when we start reading the results associated with Figure 2, no explanation has been provided in the text on the type of stressors selected and their main "categories" (antibiotic, herbicide...) or the concentrations used. Maybe provide 1-2 sentences on the rationale in the result section.

We have now grouped the chemicals by their categories, and labelled the categories in the figure (now Figure 1A). We have added a rationale for the choice of chemicals at the beginning of the results section (lines 116-118):

"These stressors are 8 different chemical pollutants known to be prevalent in freshwater environments, targeting four different components of freshwater ecosystems (bacteria, fungi, plants, and invertebrates)."

6. Quantifying multi-stressor interactions:

The approach proposed to assess stressor interactions is interesting and well explained. I just wonder why the authors only focused on interactive effects and didn't present also the results on the number of combinations that presented additive effects or no effects at all. These results are also interesting to determine the prevalence of no effect vs additivity vs antagonism/synergism and inform on the

prevalence of additive effects in stressor mixture to better predict environmental effects (Additivity is not as bad as synergism but still it is not great).

We did indeed not distinguish between additive effects (technically multiplicative in our model) and no effect. If a focal bacteria is unaffected by two chemicals and is also unaffected by combination of chemicals, this could in principle be classified as an additive effect (in essence, $0 + 0 = 0$). However, we agree with the reviewer that we can refine the method to include a separate category of 'no effect'. We have therefore updated our analyses (and updated the methods section as necessary) to now quantify combinations that produce no effects using a variation on our bootstrapping code for consistency among analyses. We have added the prevalence of "no effect" to our figures (Fig 1a,b, 4b,c,e,f, Supplementary Figure S5). We found that as more chemicals were added to mixtures, increasing proportions of those mixtures showed negative responses in their growth.

7. The panel A of figure 3 suggests that the "norm" or null hypothesis is that these stressors decrease bacterial growth and that stressors have additive effects but I wonder how often it is verified. Some stressors will have no effects at all or will increase growth. Here you display the results just based on the presence of interactive effects or not, but no info is presented on the directionality (positive/negative effects) or the proportion of treatments having no effects at all or the number of treatments presenting no interactions due to additivity (currently all presented under the "no interaction" category).

Maybe you could present or explore these other categories of responses and show some "real life" examples to describe the diversity of responses observed.

We now quantify "no effect" as a separate effect category (see comment #6, above). We show the impacts of all chemical mixtures on growth in Fig 1a, now clearly discriminating between statistically significant impacts, and those mixtures with no effect. We also now show the prevalence of positive, negative, or no effect in mixtures of different numbers of chemicals (Fig 1b), showing a small number of positive effects in mixtures of small numbers of chemicals, but increasingly negative effects as more chemicals are added to mixtures.

We have removed panel A of figure 3 (now figure 4) giving the misleading indication that the norm is to decrease growth. We retain a conceptual panel in the methods figure 6 (previously fig 5), which depicts

interactions with stressors both increasing and decreasing growth. We have updated this methods figure to show a horizontal line for “no effect”, to clarify that points above or below indicate positive or negative responses respectively.

8. For Figure 2 and Figure 3, I wonder if the analyses should not be done in presence and absence of Oxytetracycline like you did in Figure 1. We could imagine that the number of interactions might increase because of the “dominant” effect of Oxytetracycline basically killing most bacteria might hide interesting patterns between other stressors.

We now re-run the phylogenetic analyses in the absence of oxytetracycline, but still find the same lack of phylogenetic signal in the clustered responses to the chemical mixtures, these results are presented in the manuscript and supplementary figure S4. The new work that we have added on a mixed culture of isolates (see reviewer #3 comment #1) has also been analysed in the presence and absence of oxytetracycline.

We have now quantified the most prevalent interactions and find that interactions shared by many of the strains of bacteria tested here are in mixtures which all contain oxytetracycline. We now argue in our discussion that in these cases the strong impacts of oxytetracycline may actually be modulated by the addition of other chemical stressors and suggest potential mechanisms (discussion, lines 348-357):

“Interactions between antimicrobial compounds are often determined by the cellular targets of those compounds (e.g. cell wall, DNA replication machinery, ribosome synthesis, etc.), with certain target classes leading to specific interaction types with others (Bollenbach, 2015). A relevant example is bactericidal antibiotics, which require cell growth, and so compounds which inhibit growth can interact antagonistically, suppressing the effect of the antibiotic (Ocampo, 2014). Our study includes the bactericidal amoxicillin (inhibits cell wall synthesis) and bacteriostatic oxytetracycline (inhibits protein synthesis), and four strains of bacteria in our study do show an antagonism for this pairing. Moreover, the prevalence of other antagonisms observed here in mixtures with oxytetracycline may be due to bactericidal activity of pesticides, inhibiting growth and therefore reducing the efficacy of oxytetracycline.”

Note that as we have tested every possible combination of these chemicals (see our response to comment #12, below), we are able to understand the interactions in all mixtures, including those without oxytetracycline. If interactions in mixtures without oxytetracycline were prevalent, we would see them in our data, but we don't.

9. I have the feeling that some important figures have not been presented in the main text or SI. Only 5 figures are presented in the main text and 3 in SI, this looks to me that some important aspects of the datasets have not been exploited. The heatmap in figure 2 is a nice visualization but do not provide statistical tests for some conclusions related to the effect of the number of stressors (L121-122) or the similarity of responses between different strains. Simple figures showing the relationship between G and the number of stressors for each strain (each strain is a different line) or the prevalence of interactive effects for each stressor or some interesting stressor combinations could be a nice addition.

We agree with the reviewer that it is a rich dataset that could be analysed and visualised in many ways. As suggested, the heatmap now shows statistically significant responses (now Fig 1a) and we have quantified how the proportion of negative and positive impacts on growth varies with increasing numbers of chemicals (Fig 1b; see our response to comment #7 above).

We have also produced new supplementary figures showing the relationship between G and the number of stressors for each strain (Supplementary Fig S2), the most prevalent emergent interactions (Supplementary Fig S7) and an interesting suite of interactions in mixtures containing oxytetracycline and tebuconazole (Supplementary Fig S6; with further elaboration in the discussion).

We have added further statistical tests on the similarity of responses between strains through additional phylogenetic analyses (see in particular our response to reviewer #3 comment #2) and present these in a new phylogenetics figure (Fig. 3).

10. In the end, we don't really have a clear idea of the "interesting" or "surprising" interactions observed, whereas they are antagonistic or synergistic. The manuscript stays descriptive at a high level and do not provide specific examples that could provide important new knowledge in ecotoxicology.

Which stressors or types of stressors interact more frequently? In this case, which associations generate antagonistic or synergistic interactions?

In the discussion, you could add a section to discuss potential mechanisms associated with the interaction observed and illustrate it with some selected examples. How antagonism or synergism can emerge between an antibiotic and a herbicide?

We previously avoided discussion of specific interactions in favour of looking at overall patterns. However, we agree with the reviewer that some discussion of specific interactions would be a valuable addition. We address this comment with new supplementary text and figures detailing the more common interactions seen here (Supplementary figures S6 and S7), which are referred to in the results section (lines 239-245):

“Interactions in mixtures containing oxytetracycline (antibiotic) and tebuconazole (fungicide) were particularly common (Fig. 5, Supplementary Fig. S6). In higher complexity mixtures containing both oxytetracycline and tebuconazole, net interactions of the same type as the two-way interaction (antagonistic or synergistic) were also likely to be observed (Supplementary Fig. S6). Across all strains, the most prevalent higher-order emergent interactions occurred in mixtures containing both oxytetracycline and metaldehyde (Supplementary Text, Supplementary Fig. S7).”

We have added sections to the discussion to suggest potential mechanisms for these interactions (lines 348-357, see response to comment #8 above; and 357-361):

“However, there can be a multitude of additional reasons for interactions to arise, such as: uptake effects e.g., if cell permeability of one compound is affected by another (Bollenbach, 2015); if the solubility of one compound is affected by another (potentially relevant to our liquid-culture experiments); direct physical interactions, e.g., modes of action sharing the same binding site (Bollenbach, 2015); among other, more biochemically complex mechanisms.”

Specifically, we suggest that the prevalence of antagonisms with oxytetracycline may be linked to bactericidal versus bacteriostatic activity of different compounds. Oxytetracycline is bacteriostatic and

inhibits protein synthesis, and so works when cells are growing. If cell growth is inhibited by another, bactericidal compound, this may reduce the efficacy of oxytetracycline, causing the observed antagonism (see response to comment #8, above).

11. Methods:

Choice of concentration at 0.1 mg/L: I understand the rationale but why not have used specific concentrations for each stressor based on either their environmental concentrations or dose response curves (EC₅₀ or other)? Of course, the experiment cannot be redone but can you amend the table 3 with this information when they are available?

A key consideration was that we wanted to test chemical responses in mixture without their effects being impacted by differences in concentration. However, we also needed to select a concentration which had a quantifiable effect on bacterial growth, and ideally a concentration which would be ecologically relevant.

Effective concentration was a consideration, but many of the chemicals tested here do not have EC₅₀ data for bacteria, and there is the added complication that our initial experiments showed that the impacts of particular chemicals differed substantially among the strains that we used.

With those considerations in mind, we collected effective concentrations for the chemicals used here against their intended target organisms, as the ecologically relevant concentrations for these pesticides. As requested, these data are now presented in Supplementary Table S2.

We chose a concentration of 0.1mg/L, a dose that elicits a response in at least one strain of bacteria for each chemical tested (shown in supplementary figure S1). For each chemical this dose is also within an order of magnitude of the effective concentration for at least one non-bacterial taxa group, with the exception of diflufenican (EC₅₀ 0.001-0.008mg/L in algae).

12. L352: I'm not sure to understand how you obtain the number of 255 possible mixtures for the full set of combinations of 8 stressors. I would have assumed that this number will be 8 factorial $8! = 40320$. Of course, this is not feasible in a lab experiment but can you clarify the design?

The reviewer has confused permutations with combinations.

The factorial calculation suggested by the reviewer gives the number of permutations, *i.e.*, the number of ways a set can be arranged, which includes different orderings of the same elements. For example, for three chemicals A, B, and C, there are the following 3-chemical permutations:

{A,B,C}

{A,C,B}

{B,A,C}

{B,C,A}

Etc.

However, for our experiment we do not need to differentiate between mixtures containing the same chemicals listed in different orders as they all contain the same three chemicals and the order in which they are listed is not relevant.

Instead of computing the number of permutations, we need to calculate the number of combinations. For our experiment, we assayed all possible subsets of n chemicals created from the full set of 8 chemicals.

The number of combinations is 2^n (including the empty set).

For our example three-chemical set {A, B, C} we can see that there are $2^3 - 1 = 7$ possible subsets (chemical mixtures):

17{A}, {B}, {C}, {A,B}, {A,C}, {B,C}, {A,B,C},

the -1 accounts for the fact that the empty set (no chemicals at all) does not count as a mixture, rather it is the control.

Thus, for our 8 chemicals, there are $2^8 - 1 = 255$ possible chemical treatments.

Reviewer #2 (Remarks to the Author):

This manuscript on “Bacterial responses to complex mixtures of chemical pollutants” intends to address a very important and significant issue on the general topic of applied toxicology or microbial toxicology, but both the experimental design and results do not allow to advance the topic to a higher level or a significant extent in my understanding of this subject. Because of this, I feel that more specific focuses shall be given on the selection of methods, effective concentration for toxicity, mechanisms involved, the biochemical and physiological responses of the microorganisms so that the subject can benefit from the results for further advancement.

1. Microbial responses can be assessed by a arrange of determinants, but biomass by optical density is not sensitive enough because both the live and dead cells in the population were accounted without any differentiation of them, so the value is a mixture of cells of different physiological stages and also the dead ones.

We have taken bacterial biomass accumulation as our measure of the impact of the chemicals on the focal bacterial strains. Biomass accumulation (i.e., optical density, OD) is our best estimate of fitness in these systems, which would be the ultimate determinant of whether the chemicals are having interactive effects. We agree that OD measurements pool cells in different physiological states as well as dead cells, but that is a benefit and is the purpose of the assay: to get an overall measurement of how much biomass can be produced by a bacterial population under a standardised set of growth conditions

18when the only thing that is different among microcosms are the 8 chemicals. One way to think about this is that all the cells present (regardless of their current physiological state) must have been in the initial population or come into existence through successful reproduction of living cells while exposed to chemical treatments in the experiment. The initial populations were constant so any difference between mixtures must be arising from a greater or lesser amount of reproduction of bacterial cells in the presence of the relevant stressors. Indeed, our experiment does show impacts of the chemicals on our bacterial populations as measured by OD. If OD was too insensitive, as this reviewer is concerned may be the case, then we would not have seen any effects of the chemicals on OD. We therefore suggest that the OD assay is suitable for the stated purpose of our study.

2. A better and more sensitive measurement of the cells can be identified and used to achieve a more sensitive response from the metabolically active microorganisms. On the general side, heat-shock proteins and responsible genes can be used while biochemical reaction related genes can be selected for the others. In this way, the toxicity can be evaluated to exclude the metabolism involved.

The purpose of our study was not to quantify the physiological responses of the bacteria to the chemicals. If it were, we agree with the reviewer that assaying a range of genes or proteins related to stress responses would have been appropriate.

Physiological responses have metabolic costs because the resources allocated to the stress response cannot then be allocated to cell division. If the stress responses described by the reviewer are important (i.e. metabolically expensive), there would be repercussions for fitness (biomass accumulation), which is what was measured here. If the physiological responses do not have fitness consequences, they may be interesting in themselves but not important from an ecological (or indeed an ecotoxicological) perspective since there are no consequences. It is for this reason that we follow other leading publications in this field by focusing on inhibition of growth (i.e., reductions in fitness) (e.g., Maier *et. al.*, 2018).

We further note that our experiment involved measuring hundreds of chemical combinations in thousands of microcosms, and thus we required a simple assay to measure the impacts of the chemical treatments. Even a very well-funded project would have struggled to cover the costs involved in assaying the physiological state of cells across thousands of microcosms. Investigating heat shock

proteins and genes related to biochemical reactions may be more suited to a smaller-scale study focused on understanding mechanisms underlying specific interactions in higher resolution. For example, this would be an interesting follow up to some of the interactions identified here. This is an aspect of the work that is out of scope for this study, however we now mention it as potentially interesting future work in the discussion (lines 363-367):

“Further work to characterise the mode of action of pesticides on non-target organisms is therefore necessary. This may include investigating the genes related to biochemical responses and in particular should focus on whether specific classes of pesticide compounds affect microbes in similar ways and are thus generalisable.”

3. Chemicals and concentrations: Toxicology addresses the issue of chemical concentration effects on any selected organisms, but the concentration expressed and used in experiments and report shall be the active or effective concentrations, not simply the total mass per volume as targeted ones. This is very critical and has been very important to the pharmacology in medicine in evaluation of the chemical dosage to cure or kill the microorganisms for different age groups. Unfortunately, applied toxicology, ecotoxicology and environmental toxicology have been using the concentration of the environment and environmental samples too loosely by the individual involved. This is particularly true in applied and ecotoxicology when a given amount of chemicals are introduced into a testing system of marine or freshwater without checking of the soluble or effective concentration of the target chemical, and the results are registered for the supposedly chemicals for correlation with the toxic effects detected. This practice is not only wrong for the sake of science, this has generated a large data set with very low reproducibility and very high variability. Such situation prevents science for a healthy development to move to a higher level and also laying a solid foundation for future development.

Terminology can vary between pharmacology and other fields, so to avoid confusion we take the standard definition of “effective concentration” from Wikipedia (which we believe is what is meant by the reviewer here):

“In pharmacology, an effective dose (ED) or effective concentration (EC) is the minimum dose or concentration of a drug that produces a biological response.”

[https://en.wikipedia.org/wiki/Effective_dose_\(pharmacology\)](https://en.wikipedia.org/wiki/Effective_dose_(pharmacology))

The effective concentration is an important metric in toxicity testing but unfortunately has less relevance in studies like ours where we are interested in the impacts of chemicals on non-target organisms.

For example, the effective dose of a herbicide on its target plant is unlikely to tell us anything useful about its impacts on bacteria. Furthermore, these chemicals do not generally have published effective concentrations for bacteria, and even if they did, as our study shows, the effective dose varies among bacteria, further preventing us from justifying the use of a specific concentration in our study that makes comparisons between different bacteria as well as different chemical mixtures.

The solution which we implement is to move away from selecting concentrations based on biological impact, and moving toward using concentrations based on actual (or potential) environmental concentrations.

For readers that are interested in effective concentrations, we now report effective concentrations for target organisms in Supplementary Table S2, and further justify the concentration used in the methods (line 433-438; see also the response to Reviewer #1's comment #11):

“Based on preliminary work, we chose a concentration of 0.1mg/L, a dose that elicits a response in at least one strain of bacteria for each chemical tested (supplementary figure S1). For each chemical this dose is also within an order of magnitude of the effective concentration for at least one non-bacterial taxa group, based on the EPA EcoTox database (Olker et al., 2022), with the exception of Diflufenican (EC50 0.001-0.008mg/L in algae), i.e., a concentration of realistic concern to other parts of the ecosystem (Supplementary Table S2).”

This we hope will address any concerns about repeatability of the work.

Regarding soluble concentrations, we have checked the solubility of chemicals used in our experiment and now show that they are soluble at the concentration used (see our response to comment #5, below).

4. The selection of the 8 chemicals in this investigation did not seem to have any intended plan for the different chemical structures and their effects on the selected microorganisms. The justification of their widely occurrence is sound to some extent, but the results from such understanding cannot advance the in-depth science further to enhance the research subject to grow. For any chemical pollutant, there are potentially concentration of bacterial maintenance, metabolism, and then toxicity. This framework can be used to lay a good foundation for the toxicity testing and analysis. Prevalence of them is one way to justify for their relevance and significance. The chemical structures of them shall be carefully selected to yield useful information on the possible mechanism of toxicity or the gene expression so that the selective markers for monitoring can be chosen more specifically.

As the reviewer mentions, the prevalence of the chemicals in natural environments is a strong justification for the selection of pollutants for ecotoxicology studies. In addition, we specifically select the chemicals not just on their prevalence in the environment but also in terms of their mode of action, targeting 4 different ecologically important aspects of freshwater food webs (bacteria, plants, fungi and insects), and have been identified as key targets for further investigation of their impacts on microbes (Bani et al., 2022).

An alternative approach, as suggested by the reviewer, would have been to take the chemical structure of pollutants to try to infer interactive effects on the focal bacteria based on what is known about their mode of impact on bacteria. We hope that this approach will someday be possible for a study like ours. At present, however, the mode of action of most environmental pollutants on bacteria is unknown, making *a priori* inferences about interactive effects extremely challenging. Even for chemical pollutants that are well understood, interactive effects are difficult to predict - for example, there is an enormous literature on the combined impacts of antibiotics and yet bacterial physiological responses to antibiotic combinations are still poorly understood (Roemhild et al., 2022). The approach of selecting a small set of chemicals for careful study therefore contains the significant risk of missing important interactions. Our comprehensive approach of studying all possible interactions remedies this, and we think opens the door to the type of research the reviewer is most interested in, only at some future time when the prerequisite knowledge is in place.

Improved monitoring of specific chemicals may be one consequence of this work, but it is not a goal of this study so it was unclear to us why identifying selective markers would be beneficial here.

To address the reviewer concerns, we now highlight in the discussion the need to investigate whether chemicals with similar structures have similar impacts on bacteria and propose this as an avenue for future work which would help in predicting interaction effects (lines 363-367; see our response to comment #2, above).

5. The concentration of 0.1 mg/L is apparently far too high for many of the 8 chemicals to be soluble in water. Because of this, the toxicity contribution from each of the chemicals is not on an even shared basis or truly concentration determined phenomena. Furthermore, interactions of different chemicals also may result in insolubility and different effects from common exposure, additive, antagonism, mutualism and also potentiation.

We understand the concern here. However, we found that all of the chemicals used have water solubility at least an order of magnitude higher than the 0.1mg/L concentrations used. To address this comment, we now provide water solubilities of each chemical used in supplementary table S2, with source citations for each.

We do also appreciate the reviewer's comment about interactions potentially causing insolubility, this is a good point which we now add to the discussion when describing potential mechanisms for interactions (lines 359-360):

“However, there can be a multitude of additional reasons for interactions to arise, such as [...] if the solubility of one compound is affected by another (potentially relevant to our liquid-culture experiments) ...”

6. The organisms of choice: Commonly used ones are included to a large part, but *Salmonella* sp. of the Ames test for mutagenesis was not included, which is a pity because a very large database is available based on the Ames test and this study can benefit from the data and make comparison with the data available.

The Ames test is a method to determine whether a chemical compound causes DNA mutations and is therefore often employed to identify potential carcinogens. There may be a large database of these test responses, but we are unsure of how they could be linked to our experiment. While growth responses of *Salmonella* to various chemicals may have relevance to this study as a third 'model' bacteria, we do not think that mutagenesis is relevant given the timescale of our experiments. We emphasise that the aim of our study was to understand interactions in bacterial responses to *combinations* of stressors, not just the responses to single chemicals; the Ames test is generally performed on single chemicals.

There may indeed be interactions between stressors in terms of their potential to cause DNA mutations, investigating this would be interesting future work. It was not, however, the aim of our present study and does not seem to us like a natural addition to the scope.

7. The phylogeny of the selected bacteria was used for possible logical information with the toxic chemicals. This part is not likely to yield anything useful for at least two reasons. First, 16S rRNA gene is used for taxonomy and classification mostly and this gene does not contain fragments for metabolisms or adaptation. In addition, genomic DNA does not yield the activity-based information of the microorganisms, so RNA shall be used to closely monitor the metabolic activity microorganisms and gene expression.

In eco-evolutionary studies, it is commonplace to look at how traits are distributed across a phylogeny. The purpose of such studies is not, as stated by the reviewer, to quantify activity-based information, rather, the purpose is to look at whether the traits are conserved (isolated to particular branches) or not (randomly scattered across the phylogeny).

In the context of the chemicals assayed here, the response to the chemicals is the trait being measured, and the 16S locus is used solely to construct the phylogeny. If the responses to the chemicals are

conserved across the phylogeny (i.e., taxa that are more closely related have more similar responses to the chemicals), the most parsimonious explanation is that they have similar mechanisms of tolerance to the chemicals (e.g., similar metabolic responses, similar modes of action, etc). This would be because those mechanisms are inherited from a more recent common ancestor.

To test this, we need only two things: a phylogenetic tree for the species of interest, and a set of trait values to map to the tips of that tree. It is not relevant for the gene used to construct the tree to be functionally related to the phenotype being tested it just needs to be indicative of ancestry more generally. Of course, it may be that a tree built on another locus would look different, but investigating this is beyond the scope of our present work. Here we used 16S to build our bacterial phylogeny because it is standard practice in the field. We draw the reader's attention to Münkemüller et al. (2012), which gives a more thorough description around methods and applications of phylogenetic signal testing.

If the chemical responses are conserved across the phylogeny, it would greatly simplify the task of generalising the results obtained here and identifying biomarkers of chemical pollutants in ecosystems.

Investigating changes in gene expression in the responses of bacteria to these different chemical stressors would certainly be interesting, but is far outside the scope of what we were testing for here.

8. The ecology approach using fitness concept and model does not advance well on the subject matter because the competition for survival and growth is not based on resource available in this case, but the toxic chemicals on their survival and metabolism. Assuming some chemicals may be utilized by some microorganisms, then the others will need to deal with the toxic effect for survival. But the selection of the chemicals was not carefully delineated initially for the experiments, so there was not logic or plan for this part to delineate the information from the well-planned experiments.

Our starting point is that "fitness" is the currency by which the impacts of any environmental perturbation can be measured in a population, and that relative growth (measured here) is an appropriate proxy for fitness in bacteria. Here the baseline is growth in the absence of the chemical(s), and we compare that to growth in the presence of chemical(s) to obtain our measure of relative growth

(i.e., fitness). Some of the chemicals may be inhibitory to some of the bacteria, some may be metabolised (i.e., used as a resource), and others may be completely ignored. All of these potential impacts of the chemicals can be identified using our method as having a positive, negative, or no effect on growth and therefore allows us to assess the fitness consequences of the chemicals. Fitness is therefore never simply a consequence of resource competition but is rather a measure of the ability to grow and reproduce under a given set of environmental conditions.

We accept that the assays we used do not account for inter-specific interactions (like competition) since the isolates were assayed individually. We have purposefully opted for a reductionist approach that first isolates the effects of the chemicals before introducing the complexities of biotic interactions and other environmental parameters. Following reviewer comments, we have now increased the complexity of the experiments by also considering a mixture of the isolates where biotic interactions (e.g., competition) would also play a role.

9. The results in Fig. 1 are predictable for the trend and it is not new.

We retain Figure 1 (now as Fig 2a), but have added new sub-figures showing the responses of a mixed culture of strains to the chemical mixtures (fig 2b) and a comparison of the mixed culture to the mean of monoculture isolates (fig 2c). By contrasting the results of the individual strains to the mixture of strains, we hope that the reviewer finds these results more interesting and novel.

10. The chemical stressors and responses in Fig. 2 are simply presenting the data without any synthesis or conclusion can be derived from them. Such a fact reflects the serious weakness in the initial experimental design and conceptual model.

This is similar to reviewer #1's comment #9 asking for additional statistics related to these figures rather than just visualisation (see also our response to that comment above). We have now altered the heatmap figure (now Fig 1) to colour only mixtures with statistically significant responses and have added further statistical tests to our results showing the impact of oxytetracycline and the impacts of increased number of chemicals on the strength of responses. This allows us to draw more striking conclusions from our work.

11. In addition, the multi-stressors and interactions in Fig. 3 do not reveal anything of specific information and mechanism to allow a better understanding of this topic.

By adding a new category of “no response” to our results (see also reviewer 1’s comment #6) we are better able to show with this figure (now figure 4) how the prevalence of responses and interactions changes with chemical additions. We now also contrast the monoculture strain-level responses with the mixed community responses in these figures, showing that proportionally more interactions are found in monoculture than in mixture at each level of chemical mixture.

12. The interaction analysis shown in Fig. 4 is not based on a well analyzed background of these chemicals with their effective concentration so that the effects or connectivity can be interpreted.

We now highlight the most prevalent interactions and describe potential underlying mechanisms in our discussion (lines 348-361); see also our response to reviewer #1 comment #10.

We have addressed the question of effective concentrations in our response to comment #3 and now report them in the supplement.

13. The interactions of net or emergent in Fig. 5 require the concentrations of these chemicals and their toxicity must be known before making any claim of the relationship.

We have answered the point of concentration earlier (see our response to comment #5).

The “toxicity” of a chemical depends on the organism under investigation. Clearly, what is toxic to one organism may be benign to another. Our results demonstrate clearly that we can investigate interactions through statistical frameworks based on fitness without knowing toxicity in advance. There

is strong precedent for this in other microbial studies, especially for multi-drug interactions (Wood, 2012; Maier 2018, Tekin 2018).

Reviewer #3 (Remarks to the Author):

1. Broader comments

The authors conducted an ecotoxicology study on the effect of 8 stressors on the growth of 12 bacterial strains. They tried to fill two research gaps: 1) lack of studies with more than 3 stressors especially for the interactions among them and 2) most studies are based on a few model organisms. Their analytical approaches to quantify net and emergent interactions on the growth of bacterial strains are generally convincing yet some aspects and results may raise concern (see below for the specific comments). Also, the interactions at the other side of the equation should have not been ignored as well by including the mix-culture of all 12 strains as additional culture (even though it was mentioned for the future direction)

We appreciate the reviewer's comments, and agree that contrasting the monoculture responses to the response of mixtures of strains is an important addition to this study. We have therefore conducted this experiment on a mixed culture of the environmental isolates derived from a pristine system (i.e., excluding *A. fischerii* and *E. coli*). We found that in mixed culture, there is little response to the chemicals used here, and increasing the number of chemicals in mixture doesn't have the negative impact expected from monoculture experiments. We also found fewer interactions than in monoculture experiments. We hope the reviewer agrees that these are interesting and compelling new data to add to the study. We now present these results in our updated figures (Fig 1, Fig 2 and Fig 4).

We have significantly restructured our discussion to integrate these new findings, and to incorporate the reviewer's further suggestions about the responses of communities to these stressors (see our responses to comments #5, #6, #7 and #9). We have also updated the abstract to incorporate these results (lines 26-28):

"A mixed co-culture of strains proved more resilient to increasingly complex mixtures and revealed fewer interactions in the growth response."

28We have updated our methods section to describe this (lines 450-455):

“We also created a mixed culture of the pristine strains from Iceland. Here, we mixed equal volumes of the 10 strains of bacteria at carrying capacity and added this culture to the chemicals in LB such that the final mixtures contained a 1 in 100 dilution of the whole mixed culture. Therefore, in the mixed culture experiments each of the 10 strains will be at 1/10th the density of in monoculture, however the whole mixture will contain approximately the same overall quantity of bacteria as the monoculture experiments.”

2. Specific comments

L129-131: Although the Mantel test is a straightforward way to measure two dissimilarity matrices, it often is abused by overly extending from its original intent of testing dissimilarity per geographic distance. Figure 2b shows some interesting patterns that may not be well revealed by the Mantel test. There are approaches to compare two tree topologies such as comparePhylo function from ape package. The p value of 0.156 is not exceptionally high to fail to reject the null hypothesis, and I would like to see some more detailed analysis and discussions from figure 2b.

We appreciate the reviewer’s comment and tend to agree that a Mantel test is prone to type-I error and therefore may not fully reveal true phylogenetic patterns. However, we know of no better methods for testing phylogenetic signal in multiple traits simultaneously, especially where the number of traits (here, the responses to all combinations of chemicals) is much greater than the number of tips in the tree. We have looked into the comparePhylo function as suggested, but unfortunately this is not suitable for our purposes – it simply compares the topology of two trees, returning true or false depending on whether the topologies match rather than a statistical test of whether two trees differ significantly.

To address this point, we have now performed additional analyses to test for evolutionary patterns in the responses of the bacteria to each of the individual stressors alone (not the mixtures), using Pagel’s λ and Blomberg’s K, two classic metrics of phylogenetic signal. We find significant signal in the responses to amoxicillin, but not in the responses for any of the other chemicals tested, although some have “intermediate” values of the metrics, if not significant. It is perhaps unsurprising that we see signal for

29amoxicillin, given the preferential effectiveness of penicillin derivatives against gram-positive bacteria – we see this in our data with only the gram-positive *Bacillota* and *Actinomycetota* negatively impacted by amoxicillin. The results of these tests are presented in a new Figure 3b, and Supplementary Table S1.

We draw the reviewer’s attention to the fact that where they noted interesting patterns in the phylogenetics figure, this may have been in part due to the function used to draw the juxtaposed trees automatically rotating nodes to reduce the amount of cross-over of lines drawn between tips of the two trees. This leads to the two trees looking deceptively similar at first glance. We have now revised the juxtaposed trees figure (now Fig. 3a), including colour-coded taxonomic information as suggested by Reviewer #1 (comment #4), which makes the lack of phylogenetic signal more obvious.

Nevertheless, we take the reviewer’s point that this phylogenetic testing does not definitively rule out evolutionary responses to these chemicals (and indeed, we now find phylogenetic signal for one of the chemicals). Some of the lack of phylogenetic signal may also be due to the small sample size (only 12 strains of bacteria) tested. We have now softened the language around our phylogenetic findings to make clear that we are not trying to say that there is no phylogenetic component to bacterial responses to chemicals, but that phylogeny doesn’t appear to have had a strong impact here.

3. Figure 1: Linear regression results for with and without oxytetracycline in the legend are incomplete. Please include all the test statistics for both with and without oxytetracycline.

We have added the test statistics to the figure caption. Note that this is now Figure 2a in the revised manuscript.

4. L146-147 & Figure 3d & e: The conceptual distinction between net interactions and emergent interactions is clear, but I feel that there may be an alternative approach to measure emergent interactions. Figure 3e is counter-intuitive in that emergent interactions decrease as the number of stressors increases. I feel this may be an artifact from the current implementation to account for all possible subsets of chemical stressors for emergent interactions including lower-order emergent interactions, which may underestimate the emergent interactions by overestimating lower-order (emergent) interactions.

30We carefully considered this result, but we disagree that the result is counter-intuitive; in fact, we argue that this is the expected outcome. If, for instance, 8 chemicals are in mixture together, it seems intuitively more likely to us that any interaction occurring would be caused by an interaction between perhaps 2 or 3 of the chemicals within that mixture, rather than only being evident when all 8 are present. Indeed, our findings are consistent with other work in this respect, such as Wood (2012), who find that the effects of drug pairs on bacteria are often sufficient to infer the effects of larger combinations. We now cite this work in our discussion (lines 285-287):

“This is in general agreement with work on the responses of bacteria to multi-drug combinations which has established that the effects of drug pairs are often sufficient to infer the effects of larger combinations (Wood, 2012).”

To address this comment, we have also added more in-depth discussion of the potential modes of chemical interactions in our discussion (lines 348-361):

“Interactions between antimicrobial compounds are often determined by the cellular targets of those compounds (e.g. cell wall, DNA replication machinery, ribosome synthesis, etc.), with certain target classes leading to specific interaction types with others (Bollenbach, 2015). A relevant example is bactericidal antibiotics, which require cell growth, and so compounds which inhibit growth can interact antagonistically, suppressing the effect of the antibiotic (Ocampo, 2014). Our study includes the bactericidal amoxicillin (inhibits cell wall synthesis) and bacteriostatic oxytetracycline (inhibits protein synthesis), and four strains of bacteria in our study do show an antagonism for this pairing. Moreover, the prevalence of other antagonisms observed here in mixtures with oxytetracycline may be due to bactericidal activity of pesticides, inhibiting growth and therefore reducing the efficacy of oxytetracycline. However, there can be a multitude of additional reasons for interactions to arise, such as: uptake effects e.g., if cell permeability of one compound is affected by another (Bollenbach, 2015); if the solubility of one compound is affected by another (potentially relevant to our liquid-culture experiments); direct physical interactions, e.g., modes of action sharing the same binding site (Bollenbach, 2015); among other, more biochemically complex mechanisms.”

We hope that the reviewer agrees that these mechanisms are more likely to come into play with few chemicals, rather than requiring complex mixtures of many chemical compounds in order to occur.

5. L213: Yes, but the important step in between should be on the response as a community of microbes for the biological interactions, since they have very important role in ecosystem processes.

We have now added new results on a mixed culture of isolates, which we relate to the implications for communities in further parts of the discussion (see response to comment #9, below).

6. L227-229: Using “similar” prevalence of antagonistic and synergistic responses as evidence to use microbes for macro-organisms reminds me of the old ecological research trend of creating microbial “black box” since it doesn’t account for the apparent distinction of physiology between them. However, I can agree with the authors that microbes can be great bioindicators for ecotoxicological studies, but the reason is because of the significant role of microbes in ecosystem stability and function.

The reviewer makes a good point here. We have removed the lines with the “black-box” implication and added a new point around microbial biosensors (line 342-346):

“While this suggests a limited use for individual taxa as indicators of chemical impacts on the wider community, it doesn’t rule out their use in other contexts. Indeed, the wide variation in responses of different bacteria to chemicals seen here may support the use certain microbial taxa as bioindicators for the entry of specific chemicals into natural environments.”

7. L231-233: Another good reason the biotic interactions among bacterial strains should be considered for a more comprehensive understanding here.

We agree, and have integrated the results from a mixed culture of isolates into our discussion (see response to comment #9, below). Additionally, we have moved the paragraphs on communities and biotic interactions to earlier in the discussion to increase their prominence.

8. L241-243: I'd like to see more reasoning on how insignificant associations between phylogenetic and phenotypic distribution support convergent evolution on physiological responses.

Phylogenetic conservatism suggests that traits arise in common ancestors and get shared between close relatives. Lack of such conservatism thus suggests that traits arise independently more than once (convergent evolution). However, in softening the language around our phylogenetic findings (see response to comment #2), we have opted to remove these lines from the manuscript as the argument is not overly relevant to the point that we were trying to get across.

9. L252-271: I appreciate the proposed future work focusing on microbial communities. However, or because of this, I can't really agree with the implications the authors try to make based on the relationship between species richness and multifunctionality. I think it is an overly stretched point and removing it may make the latter half of the paragraph more convincing.

We have significantly reworked this paragraph, in part due to our addition of new data. We have removed the section on species richness and multifunctionality as suggested. We have now linked our own findings of fewer interactions and lesser response of the mixed culture, to potential resilience of community functioning (lines 299-303):

“We found that the growth of a mixed culture of strains was both less impacted by the addition of multiple chemicals and showed many fewer interactions than expected from the growth of the strains in monoculture. This shows that when chemicals inhibit part of the community, compensation by resistant taxa may rescue important functions, such that growth is maintained.”

We have also highlighted the importance of keystone species, despite the potential for overall resilience of communities (lines 303-309):

“However, while the growth of microbial communities may be resilient to multiple stressors in a broad sense, interactions may still be impactful in natural communities. In particular, keystone taxa can have an over-proportional impact in driving the composition and functioning of microbial communities (Berry, 2014; Banerjee, 2018). At a single strain level, we found highly diverse responses to the chemical mixtures. If unique interactions lead to the loss of keystone species, this could have comparatively large effects on community functioning and stability.”

10. L377-381: The use of a multiplicative null model need more explanation. It may not be just me having a hard time being convinced why the combined effects based on relative fitness measures must be the products, not the sum. Cited work by Tekin et al. 2018 in npj doesn't seem to support the use of the multiplicative model, as their null model of multiway interactions is based on an additive model.

We use a multiplicative model as using the sum instead of the products can lead to biologically meaningless predictions when using relative fitness. For example, if two stressors each reduce bacterial growth by 70%, then an additive model would predict their combination to reduce growth by 140%. This is not meaningful as a reduction in growth must be bounded at 100%, i.e. zero measurable growth. Using a multiplicative model, we would predict a 91% reduction in growth in this example $(1-(1-0.7)^2)$. We find this to be a more biologically meaningful and intuitive model. We have added this justification to the methods section to clarify the model choice (lines 485-493):

“We considered multiplicative, and not additive null models because we measured relative fitness, which is equivalent to a percentage change in growth. Using an additive model for relative fitness could lead to biologically meaningless predictions. For example, if two stressors each reduce bacterial growth by 70%, then an additive model would predict their combination to reduce growth by 140%. This is not meaningful as a reduction in growth must be bounded at 100%, i.e. zero measurable growth. Using a multiplicative model, we would predict a 91% reduction in growth in this example $(1-(1-0.7)^2 = 0.91)$. Therefore, the null model for the combined effects must be the product of the percentages, not the sum (Tekin, White, et al., 2018).”

In the cited work by Tekin et al, they do use the term “additive”, however mathematically their model is technically multiplicative. This is the relevant section from Tekin et al. (2018), with the description of a null multiplicative model highlighted in bold:

*“For drug studies the key response measurement is growth rate of a bacteria population in the presence of a drug relative to growth of bacteria in a no-drug environment. This growth rate is interpreted as the relative fitness in the presence of a drug treatment X and is typically denoted by w_x (Fig. 1), where no- growth ($w_x= 0$) represents complete lethality and maximum-growth ($w_x= 1$) represents the case that the drug treatment is not effective at all. **Because we are using relative fitness, the effect of each individual drug can be interpreted as a percent reduction in growth rate, so the null expectation for the combined effects of two non-interacting drugs would be the product of two percentages, corresponding to a multiplicative definition of no interaction. ... Consistent with the drug literature, we will use the term additivity to refer to no interaction throughout the rest of this paper.**”*

Therefore, Tekin et al. use the term “additive” to describe their null model, but this is not mathematically accurate, the model is multiplicate. This is also the case for their other papers on multiple stressor testing (e.g., Tekin et al. 2016; 2020), where the same multiplicative model is used, but additive is the term used. We think that this is likely to cause confusion to readers, and so prefer to be explicit in our paper about the null model being multiplicative.

11. L442-450: Does any of the null expectation was positive with those stressors? I think including positive null expectation is unnecessary and may cause confusion to some readers. Both figures 3a and 5 are based on negative null expectations only.

There are some significantly positive responses to these chemicals (though generally smaller in magnitude than the more abundant negative responses, see our revised Figure 1), leading to a small number of positive null expectations, and so we feel this is an important consideration to retain. We have removed the conceptual panel in Figure 3 (now Figure 4 in the revised manuscript), but retain the conceptual Figure 5 (now Figure 6). While this figure is based on a negative null expectation, it includes a positive response for one of the chemicals (now more clearly marked).

12. L468: Do this phenotypic data refer to the growth response to stressors?

Yes, we have now clarified this in the text (lines 588-589).

13. L474: Report the version of vegan package as you did for R on line 452.

We now provide a version number and citation for the *vegan* package, as well as the *Phytools* package which we have used for additional phylogenetic analyses.

References

- Bani, A., Randall, K. C., Clark, D. R., Gregson, B. H., Henderson, D. K., Losty, E. C., & Ferguson, R. M. W. (2022). Mind the gaps: What do we know about how multiple chemical stressors impact freshwater aquatic microbiomes? *Advances in Ecological Research*, 67, 331-377.
- Maier, L., Pruteanu, M., Kuhn, M., Zeller, G., Telzerow, A., Anderson, E. E., Brochado, A. R., Fernandez, K. C., Dose, H., Mori, H., Patil, K. R., Bork, P. & Typas, A. (2018) Extensive impact of non-antibiotic drugs on human gut bacteria. *Nature*, 555, 623-628.
- Münkemüller, T., Lavergne, S., Bzeznik, B., Dray, S., Jombart, T., Schiffrers, K. & Thuiller, W. (2012). How to measure and test phylogenetic signal. *Methods in Ecology and Evolution*, 3, 743-756.
- Olker, J. H., Elonen, C. M., Pilli, A., Anderson, A., Kinziger, B., Erickson, S., Skopinski, M., Pomplun, A., LaLone, C. A., Russom, C. L. & Hoff, D. (2022) The ECOTOXicology Knowledgebase: A Curated Database of Ecologically Relevant Toxicity Tests to Support Environmental Research and Risk Assessment. *Environmental Toxicology and Chemistry*, 41(6), 1520-1539.
- Roemhild, R., Bollenbach, T. & Andersson, T. I. (2022) The physiology and genetics of bacterial responses to antibiotic combinations. *Nature Reviews Microbiology*, 20, 478-490.

- Tekin, E., Beppler, C., White, C., Mao, Z., Savage, V. M., & Yeh, P. J. (2016). Enhanced identification of synergistic and antagonistic emergent interactions among three or more drugs. *Journal of the Royal Society Interface*, 13(119), 18–20. <https://doi.org/10.1098/rsif.2016.0332>
- Tekin, E., White, C., Kang, T. M., Singh, N., Cruz-Loya, M., Damoiseaux, R., Savage, V. M., & Yeh, P. J. (2018). Prevalence and patterns of higher-order drug interactions in *Escherichia coli*. *Npj Systems Biology and Applications*, 4(1). <https://doi.org/10.1038/s41540-018-0069-9>
- Tekin, E., Diamant, E. S., Cruz-Loya, M., Enriquez, V., Singh, N., Savage, V. M., & Yeh, P. J. (2020). Using a newly introduced framework to measure ecological stressor interactions. *Ecology Letters*, 1391–1403.
- Wood, K., Nishida, S., Sontag, E. D. & Cluzel, P. (2012). Mechanism-independent method for predicting response to multidrug combinations in bacteria. *Proceedings of the National Academy of Sciences of the United States of America*, 109(30) 122254-122259.

Decision Letter, first revision:

Message: Our ref: NMICROBIOL-22102628B

11th January 2024

Dear Dr. Smith,

Thank you for your patience as we've prepared the guidelines for final submission of your Nature Microbiology manuscript, "Bacterial responses to complex mixtures of chemical pollutants" (NMICROBIOL-22102628B). Please carefully follow the step-by-step instructions provided in the attached file, and add a response in each row of the table to indicate the changes that you have made. Please also check and comment on any additional marked-up edits we have proposed within the text. Ensuring that each point is addressed will help to ensure that your revised manuscript can be swiftly handed over to our production team.

If you have not done so already, please alert us to any related manuscripts from your group that are under consideration or in press at other journals, or are being written up for

37submission to other journals (see: <https://www.nature.com/nature-research/editorial-policies/plagiarism#policy-on-duplicate-publication> for details).

In recognition of the time and expertise our reviewers provide to Nature Microbiology's editorial process, we would like to formally acknowledge their contribution to the external peer review of your manuscript entitled "Bacterial responses to complex mixtures of chemical pollutants". For those reviewers who give their assent, we will be publishing their names alongside the published article.

Nature Microbiology offers a Transparent Peer Review option for new original research manuscripts submitted after December 1st, 2019. As part of this initiative, we encourage our authors to support increased transparency into the peer review process by agreeing to have the reviewer comments, author rebuttal letters, and editorial decision letters published as a Supplementary item. When you submit your final files please clearly state in your cover letter whether or not you would like to participate in this initiative. Please note that failure to state your preference will result in delays in accepting your manuscript for publication.

Cover suggestions

COVER ARTWORK: We welcome submissions of artwork for consideration for our cover. For more information, please see our [guide for cover artwork](https://www.nature.com/documents/Nature_covers_author_guide.pdf).

Nature Microbiology has now transitioned to a unified Rights Collection system which will allow our Author Services team to quickly and easily collect the rights and permissions required to publish your work. Approximately 10 days after your paper is formally accepted, you will receive an email in providing you with a link to complete the grant of rights. If your paper is eligible for Open Access, our Author Services team will also be in touch regarding any additional information that may be required to arrange payment for your article.

Please note that *Nature Microbiology* is a Transformative Journal (TJ). Authors may publish their research with us through the traditional subscription access route or make their paper immediately open access through payment of an article-processing charge (APC). Authors will not be required to make a final decision about access to their article until it has been accepted. [Find out more about Transformative Journals](https://www.springernature.com/gp/open-research/transformative-journals)

Authors may need to take specific actions to achieve [compliance with funder and institutional open access mandates](https://www.springernature.com/gp/open-research/funding/policy-compliance-faqs). If your research is supported by a funder that requires immediate open access

(e.g. according to [Plan S principles](https://www.springernature.com/gp/open-research/plan-s-compliance)) then you should select the gold OA route, and we will direct you to the compliant route where possible. For authors selecting the subscription publication route, the journal's standard licensing terms will need to be accepted, including [self-archiving policies](https://www.nature.com/nature-portfolio/editorial-policies/self-archiving-and-license-to-publish). Those licensing terms will supersede any other terms that the author or any third party may assert apply to any version of the manuscript.

Reviewer #1:

Remarks to the Author:

I thank the authors for their detailed responses to my comments. They included many new figures and information in the text, but also new interesting results based on a synthetic community. The new figures 1,2,3 provide very impactful and novel results.

The discussion was also significantly revised.

I have no further comments and congratulate the authors for their important contribution to the field

Reviewer #2:

Remarks to the Author:

This manuscript on "Bacterial responses to complex mixtures of chemical pollutants" has been revised to some extents, but I feel the contents are still weak for a journal of this level of publicity to serve the international scientific community. Issues include the selection of the methods, effective/bioavailable concentration of toxicity, mechanisms involved, the biochemical responses of the microorganisms so that the subject can benefit from the results for further advancement. For an example of specific, addition of phylogenetic analyses to show does not enhance the scientific quality of this manuscript significantly because you had reconstituted a consortium for the test and the microorganisms are all there when using DNA for the analysis. However, if RNA was used, your results will be guaranteed different and more explanation will be generated accordingly. On the interactions between Oxytetracycline and Tebuconazole, what are the underlying science for a non-descriptive information?

391) When more sensitive scientific methods are available, it is necessary to use the industrial manufacturing for sure. For science, specificity and precision are the key factors to striving for convincing and new results. It is for this purpose I asked for new and more innovative techniques to be used.

2) Chemical concentration shall be on the effective/bioavailable basis, this is new and I do not think many of the biologists or environmental scientists can take this seriously at the moment. To discover the basis of the responses of stressors, the concentration term shall be on a common meaningful basis and this is why I emphasize on the effective or bioavailable fractions or concentration here. Pharmacology needs to worry much less or any for this, but ecotoxicology and environmental toxicology cannot take this lightly to avoid falling into the trap of weak science.

3) If 'fitness' is the currency for the system analysed here as you claimed, then the history and evolutionary changes shall be analyzed with more robust methods. If not, it is the hypothesis, not the data to support your description.

Reviewer #3:

Remarks to the Author:

The reviewer appreciates the efforts the authors put in additional experiments, analyses, and manuscript revision. I very much agree and accept all the responses to my comments.

Author Rebuttal, first revision:

Reviewers #1 and #3 felt their comments had been addressed and had no further comments. We address the Reviewer #2 comments below. The reviewer's comments are in black, our responses are in blue.

Reviewer #2 (Remarks to the Author):

This manuscript on “Bacterial responses to complex mixtures of chemical pollutants” has been revised to some extents, but I feel the contents are still weak for a journal of this level of publicity to serve the international scientific community. Issues include the selection of the methods, effective/bioavailable concentration of toxicity, mechanisms involved, the biochemical responses of the microorganisms so that the subject can benefit from the results for further advancement.

We have made extensive changes to our manuscript based on the reviewer's previous comments. We hope that we can convince the reviewer that we have addressed their remaining concerns in the comments below.

40For an example of specific, addition of phylogenetic analyses to show does not enhance the scientific quality of this manuscript significantly because you had reconstituted a consortium for the test and the microorganisms are all there when using DNA for the analysis. However, if RNA was used, your results will be guaranteed different and more explanation will be generated accordingly.

We think that the reviewer might be misunderstanding the purpose of these analyses. The phylogenetic analysis, and analysis of chemical impacts on the reconstituted consortium were two separate investigations intended to answer different questions. The phylogenetic analysis asked whether evolutionarily related strains of bacteria elicited similar responses when exposed to the chemical treatments in monoculture; these tests were not performed on the consortium, as the reviewer implies. The analysis of the consortium of bacteria was to identify whether biotic interactions had an impact on the responses of a community of bacteria exposed to the chemical treatments. These are separate questions.

We agree that performing analyses on RNA instead of DNA may have provided different results – specifically, we could ask whether gene expression changes in response to the chemicals. However, this is a different question to what we were aiming to answer, which was specifically geared towards identifying whether phylogenetic relatedness (i.e. genomic DNA) could be used to predict impacts of the chemicals. We note the reviewer’s interest in RNA analyses, and have added this to our discussion by now specifying that understanding changes in gene expression would be a useful avenue for future investigation (lines 282-285):

“Further work to characterise the mode of action of pesticides on non-target organisms is therefore necessary. This may include investigating the expression of genes related to biochemical responses and in particular should focus on whether specific classes of pesticide compounds affect microbes in similar ways and are thus generalisable”.

On the interactions between Oxytetracycline and Tebuconazole, what are the underlying science for a non-descriptive information?

We have suggested a potential broad mechanism for this interaction which may be due to bactericidal activity of the pesticide coupled with the bacteriostatic activity of oxytetracycline (lines 271-278). However, we recognise that the underlying mechanism is unknown, and more research is needed. We now emphasise throughout the Discussion that identifying the

mechanistic basis for the observed interactions would be an obvious next step (e.g. lines 286-288):

“Further work on the mechanistic basis for the interactions we observed would lay the foundation for future ecotoxicology frameworks that predict the impacts of multiple chemical compounds on the non-target microbial components of ecosystems more effectively”.

1) When more sensitive scientific methods are available, it is necessary to use the industrial manufacturing for sure. For science, specificity and precision are the key factors to striving for convincing and new results. It is for this purpose I asked for new and more innovative techniques to be used.

We agree that specificity and precision are key factors in striving for new results. This is why we have performed our high throughput experiment using a pipetting robot to create the chemical mixtures, and a stacking plate reader to automatically assay growth in hundreds of microcosms simultaneously. This minimises the possibility of human error in our experiments, enhancing precision and thus confidence in our findings. The underlying methods are standard in the field (e.g. Maier et al.¹, Tekin et al.²), but the innovations in automation used here allow us to perform these experiments at an unprecedented level of throughput.

2) Chemical concentration shall be on the effective/bioavailable basis, this is new and I do not think many of the biologists or environmental scientists can take this seriously at the moment. To discover the basis of the responses of stressors, the concentration term shall be on a common meaningful basis and this is why I emphasize on the effective or bioavailable fractions or concentration here. Pharmacology needs to worry much less or any for this, but ecotoxicology and environmental toxicology cannot take this lightly to avoid falling into the trap of weak science.

We agree with the reviewer that bioavailability may play a role in the effect of certain chemicals on organisms - if they are not bioavailable, a higher concentration will be required to see effects. Indeed, this might lead to the different responses seen in different bacteria. As such, we have now added discussion of bioavailability to our manuscript (lines 261-262):

“Bioavailability also plays a key role in this context, because measured concentrations may not reflect the capacity for entry into target organisms”.

However, here we were not aiming to uncover the specific mechanisms of toxicity of certain chemicals, but the potential for interactions in bacterial responses to multiple chemicals. We have reported the concentration used and provide a rationale for why we used that specific concentration. We believe that this is sufficient to make our conclusions - the bioavailability is a separate question that is best dealt with in a separate study. We reiterate that we are not trying to discover the full mechanistic bases of stressor responses in this paper, but more general patterns in the types of interactions present.

3) If 'fitness' is the currency for the system analysed here as you claimed, then the history and evolutionary changes shall be analyzed with more robust methods. If not, it is the hypothesis, not the data to support your description.

We have improved our phylogenetic analyses in response to the previous round of revisions. By testing for phylogenetic signal in the responses of bacteria to the chemical mixtures (measured as a change in "fitness"), we are analysing whether evolutionary history plays a role in the differing responses of different bacteria. The analyses presented here do indeed allow a robust test of these ideas, which have been confirmed by the other reviewers.

1. Maier, L. *et al.* Extensive impact of non-antibiotic drugs on human gut bacteria. *Nature* **555**, 623–628 (2018).
2. Tekin, E. *et al.* Prevalence and patterns of higher-order drug interactions in *Escherichia coli*. *NPJ Syst Biol Appl* **4**, (2018).

Final Decision Letter:

Message 30th January 2024

:
Dear Dr Smith,

I am pleased to accept your Article "High throughput characterization of bacterial responses to complex mixtures of chemical pollutants" for publication in *Nature Microbiology*. Thank you for having chosen to submit your work to us and many congratulations.

Over the next few weeks, your paper will be copyedited to ensure that it conforms to *Nature Microbiology* style. We look particularly carefully at the titles of all papers to ensure that they are relatively brief and understandable.

Once your paper is typeset, you will receive an email with a link to choose the appropriate

43publishing options for your paper and our Author Services team will be in touch regarding any additional information that may be required. Once your paper has been scheduled for online publication, the Nature press office will be in touch to confirm the details.

You may wish to make your media relations office aware of your accepted publication, in case they consider it appropriate to organize some internal or external publicity. Once your paper has been scheduled you will receive an email confirming the publication details. This is normally 3-4 working days in advance of publication. If you need additional notice of the date and time of publication, please let the production team know when you receive the proof of your article to ensure there is sufficient time to coordinate. Further information on our embargo policies can be found here:

<https://www.nature.com/authors/policies/embargo.html>

Please note that *Nature Microbiology* is a Transformative Journal (TJ). Authors may publish their research with us through the traditional subscription access route or make their paper immediately open access through payment of an article-processing charge (APC). Authors will not be required to make a final decision about access to their article until it has been accepted. [Find out more about Transformative Journals](https://www.springernature.com/gp/open-research/transformative-journals)

Authors may need to take specific actions to achieve [compliance with funder and institutional open access mandates](https://www.springernature.com/gp/open-research/funding/policy-compliance-faqs). If your research is supported by a funder that requires immediate open access (e.g. according to [Plan S principles](https://www.springernature.com/gp/open-research/plan-s-compliance)) then you should select the gold OA route, and we will direct you to the compliant route where possible. For authors selecting the subscription publication route, the journal's standard licensing terms will need to be accepted, including [self-archiving policies](https://www.nature.com/nature-portfolio/editorial-policies/self-archiving-and-license-to-publish). Those licensing terms will supersede any other terms that the author or any third party may assert apply to any version of the manuscript.

With kind regards,